



# Multi-cooperation of soil biota in the plough layer is the key for conservation tillage to improve N availability and crop yield

Shixiu Zhang[a], Liang Chang[a], Neil B. McLaughlin[b], Shuyan Cui[c,d], Haitao Wu[a, *], Donghui Wu[a], Wenju Liang[c], Aizhen Liang[a, *]

[a] Key Laboratory of Mollisols Agroecology, Northeast Institute of Geography and Agroecology, Chinese Academy of Sciences, Changchun 130012, China

[b] Ottawa Research and Development Centre, Agriculture and Agri-Food Canada, Ottawa, K1A 0C6, Canada

[c] Institute of Applied Ecology, Chinese Academy of Science, Shenyang 110016, China

[d] Liaoning Normal University, Liaoning 110036, China

*Correspondence to*: Haitao Wu (wuhaitao@iga.ac.cn) or Aizhen Liang (liangaizhen@iga.ac.cn)

## Abstract

Conservation tillage facilitates constructing a more complex and heterogeneous distribution of soil organisms in the plough layer relative to conventional tillage (CT), which results in an improvement in crop yield and nitrogen (N)

uptake. However, knowledge of how soil biota interact to couple mineralization of N and promote plant growth is still limited. The contribution of soil biota (trophic groups and energy pathways) to soil N mineralization and the relationship between energy pathways and grain yield during soybean (Glycine max Merr.) growing season were investigated at 0–5 and 5–15 cm depths under a long-term tillage trial. The trial was initiated in 2001 on a Black soil in Northeastern China and included no tillage (NT), ridge tillage (RT) and CT. A higher contribution of most trophic

groups to soil N mineralization throughout the whole plough layer was observed in RT and NT than in CT, and these differences were more pronounced for higher trophic groups than for lower ones. Furthermore, the responses of trophic groups to tillage practices were also transferred into the energy pathways. Bacterial and predator-prey pathways released more mineral N in RT and NT than in CT. Multiple regression models revealed that soybean yield was significantly related to the mineralized N in RT and NT through root, fungal and prey-predator pathways in 0–5 cm and

bacterial pathway in 5–15 cm. Additionally, the relative contribution of the mineralized N by different pathways to soybean yield was different in 0–5 cm and decreased in the order of root pathway (0.487) > fungal pathway (0.389) > predator-prey pathway (0.318). Although soil trophic groups coupled with N mineralization and soybean yield varied with depth in RT and NT soils, a stable supply of mineral N from soil to plant could be maintained in the plough layer by the cooperation of predator-prey pathway horizontally with bacterial and plant pathways and vertically with fungal

and bacterial pathways. This favorable effect of multi-cooperation of soil biota on coupling N mineralization and plant growth in the plough layer is a cornerstone of conservation tillage benefits in temperate areas of the world.



## 1. Introduction

Crop yield is the ultimate goal of agriculture practices. A high crop yield is usually achieved by increasing nitrogen (N) availability for plants since N is the most important growth-limiting nutrient (Bardgett and Chan, 1999; van Groenigen et al., 2015). Organic form is the primary form of N stored in soil, which can only be available to plants after mineralization by soil biota. Numerous reports (Cole et al., 2002; Rashid et al., 2014; van Groenigen et al., 2015) have documented that the presence of soil biota increases N mineralization, accounting for 30–80% of the total N

mineralization of soil (Schon et al., 2012; Ruiter et al., 1993; Hunt et al., 1987). However, these cited studies often focus on only one trophic interaction, e.g. direct microbial mineralization or fauna indirectly affecting N mineralization (Brussaard et al., 2007; van Groenigen et al., 2015), and rarely consider their interactions. Indeed, soil biota is an integral component in an ecosystem. The interactions of soil biota not only lead to the changes in the composition of special species associated with the N mineralization and immobilization, but also act as linkages to coupling the N

flow from different trophic groups in soil food web (Brussaard et al., 2007; Holtkamp et al., 2011; van Groenigen et al., 2015). Therefore, soil biotic interactions are at the heart in mediating the rate of N mineralization and its availability for plants.

Separating soil food webs into different pathways is useful for aiding our understanding of the effects of soil biota on N availability to plants, as these pathways represent distinctive trophic interactions (plant-herbivore,

microflora-microfauna, and soil fauna-fauna) and have unique roles in N mineralization (de Vries et al., 2013; Holtkamp et al., 2011; Rooney and McCann, 2012). For example, weak root infections of herbivorous nematodes may increase N availability to plants (Bonkowski, 2009). Microflora and microfauna interactions can be further divided by bacterial and fungal pathways, which have contrasting mechanisms on N mineralization with fast vs. slow turnover rates (de Vries et al., 2012; Moore et al., 2005). Although these pathways exhibit different roles in N mineralization,

their capacity in N mineralization is closely related with each other. The broadly held view is that the interaction between plant and herbivores (i.e. plant pathway) fuel the nutrient mineralization through bacterial and fungal pathways (Wall et al., 2015). Soil faunal interaction, indicated by the predator-prey pathway, can shift energy flow from one pathway to another, thus increasing the stability of soil nutrient supply to plants (Moore et al., 2005).

It is increasingly recognized that soil biotic traits can be exploited through appropriate management to obtain

sustainable productivity (Wall et al., 2015). Conservation tillage, adopted worldwide to conquer the adverse effects of conventional tillage (CT), has positive effects on soil biota (Wall et al., 2015). The improvement of biodiversity and the enhancement of trophic group interactions in conservation tillage are expected to cause parallel changes in

mediating N availability to plants. Several reports (Bardgett and Chan, 1999; Cole et al., 2002; Postma-Blaauw et al., 2006) based on inoculation for increasing the abundance of specific species and/or enhancing the interaction of trophic

groups provide strong evidence that conservation tillage may improve N mineralization and subsequent plant uptake because of its rich and abundant soil biota. Nevertheless, this assumption needs to be verified in the field, as these cited reports were conducted in controlled micro- or meso-cosms where conditions differ from the natural field conditions. Furthermore, stratification is a typical characteristic of conservation tillage due to the fact that plant residues are left on the soil surface (Ehlers and Claupein, 1994). This stratification occurs not only in the physical and chemical properties

but also in biological communities, which is in contrast to the relatively uniform vertical distribution of soil biota in the plough layer of CT. Bacteria and bacterivorous fauna are dominant in the plough layer of CT, while conservation tillage is characterized by the fungi and fungivorous fauna near the surface and a bacterial-based food web at greater depths (Hendrix et al., 1986; van Capelle et al., 2012). Obviously, this heterogeneous composition of soil community in the plough layer confers different effects on N mineralization. Moreover, the enhanced abundance of herbivorous

and predaceous fauna in conservation tillage also contributes to multi-trophic interactions, further complicating N mineralization in the soil food web. However, our understanding of which groups are involved in N mineralization in different soil layers, and how the spatially separated organisms interact to affect N mineralization and plant growth remains unclear.

The objective of this study was to investigate the effect of soil biota on soil N mineralization and its contribution to

plant yield under contrasting tillage practices in a long-term (15 years) tillage trial. Soil biota examined included microbes, nematodes, mites and collembolans, and were classified into four main energy pathways, accounting for the trophic interactions and the N flowing through the food web (de Vries et al., 2012; Holtkamp et al., 2011). We hypothesized that (1) conservation tillage increases the contributions of soil biota to coupling soil N mineralization and plant yield, (2) conservation tillage favors multiple spatial interactions of soil biota through different energy pathways

in the plough layer to provide a stable N supply for plant growth.

## 2 Material and methods

### 2.1 Experimental design and soil sampling

This study was conducted at the Experimental Station (44°12'N, 125°33'E) of the Northeast Institute of Geography and

Agroecology, Chinese Academy of Sciences, in Dehui County, Jilin Province, China. The station is located in a continental temperate monsoon zone. The soil is classified as Black soil (Typic Hapludoll, USDA Soil Taxonomy)



with a clay loam texture. Tillage experiment was established in the fall of 2001 and included conventional tillage (CT), ridge tillage (RT) and no tillage (NT) in a two year maize (Zea mays L.) - soybean (Glycine max Merr.) rotation system. Each treatment had four replications.

The 2-yr maize-soybean rotation with residue return was considered in this study. Details of the experiment layout, tillage applications, crop rotations and fertilization were reported by Zhang et al. (2019). Briefly, CT practice consisted of fall mouldboard ploughing (20 cm) followed by the secondary seedbed preparation in the spring by disking (7.5–10 cm), harrowing and ridge-building. In RT, ridges were formed with a modified lister and scrubber and were maintained in June of each year with a cultivator. For the NT, no soil disturbance was practiced except for planting using a no-till

planter. After harvest, the maize residue in the RT and NT plots was cut into about 30 cm pieces leaving a 30–35 cm standing stubble; soybean residue was directly returned to the soil surface. Residues in CT plots were removed prior to, and manually replaced on the soil surface after fall mouldboard ploughing.

Soil samples were taken at the end of each month from April to September 2015 during the soybean growing season when soil organisms are active. Seven soil cores (2.5 cm in diameter) in each plot were randomly collected from a

depth of 15 cm and each core was separated into 0–5 and 5–15 cm sections. Soil cores were combined to form a single composite sample for each plot and depth. Samples were immediately taken to the lab and stored at 4 °C. Soil temperature was recorded with a data logger (Watchdog, Model 1525, Spectrum Technologies, Aurora, IL USA) and soil bulk density was determined at 5 cm and 10 cm. The mean monthly bulk density and soil temperature are presented in supplementary Table 1 (hereafter 'S' is used as the abbreviation of supplementary).


**2.2 Soil mineral N and soybean yield analysis**

Soil N mineralized (SNM) during the soybean growing season is the sum of the monthly potential N mineralization from April to September and corrected by field temperature:

$$SNM = \sum_t Q_{10}^{(ST-T_0)/10} \times Np_t \qquad (1)$$

Where t indicates the specific sampling month; ST is the monthly mean soil temperature; T0 is the reference temperature at which potential mineralization was determined (20°C); the $Q_{10}$ is assumed to be 3 (Bloem et al., 1994); $Np_t$ is the potential N mineralization at the t specific month (mg N (kg soil)$^{-1}$ (4 weeks)$^{-1}$). Np was determined by the difference of mineral N content before and after incubation of 20 g field-moist soils in the dark at temperature $T_0$ (20°C) for 4 weeks (Chen et al., 2019). Mineral N was extracted with 2 M KCl (1:5 wt/vol, shaken at 160 cycles min$^{-1}$ for 1

hour), and measured using a flow injection auto analyzer (FIAstar 5000 Analyzer; Foss Tecator, Hillerød, Denmark).





SNM was up-scaled to kg ha$^{-1}$ using soil bulk density and depth of soil sample (5 or 10 cm).

Soybean yield was determined by hand-harvesting 3 m lengths of 6 interior rows from each plot after plants had reached the physiological maturity. Grain yield samples were dried to a constant weight at 75 °C in an oven, and then corrected to 13.5% grain moisture content.


## 2.3 Soil organism extraction

Soil biota were extracted within 2 weeks of sample collection to ensure the maximum activity. All soil biota were determined monthly except nematodes, which were only determined in April, June and August due to the limitation of labor. The nematode populations for non-sampled months were estimated by linear interpolation.

Microbes were extracted from 8 g of freeze-dried soil with a Bligh and Dyer solution (chloroform: methanol: citrate buffer = 1: 2: 0.8 (v: v: v)) using a phospholipid fatty acids analysis as described by Bossio and Scow (1998). Lipids were separated from neutral lipids and glycolipids in a solid phase extraction column (Supelco Inc., Bellefonte, PA, USA) and transformed into fatty acid methyl esters with a mild alkaline methanolysis. Samples were then dissolved in hexane and analyzed in an Agilent 6850 series Gas Chromatograph with MIDI peak identification software (Version 135 4.5; MIDI Inc., Newark, DE, USA). Fatty acids were grouped as bacteria (14:0, i14:0, a14:0, 15:0, i15:0, a15:0, 15:1ω6c, 16:0, i16:0, a16:0, 16:1ω7c, 16:1ω9c, i17:0, a17:0, 17:1ω8c, 17:1ω9c,18:1ω7c, 18:0, 20:0), saprophytic fungi (18:1ω9c and 18:2ω6c) and arbuscular mycorrhizal fungi (AMF) (16:1ω5c) (Bach et al., 2010; Dempsey et al., 2013). Microbial biomass was estimated using the following conversion factors of fatty acid concentrations (nmol): bacterial biomass, 363.6 nmol = 1 mg C; saprophytic fungal biomass, 11.8 nmol = 1 mg C; and AMF biomass, 1.047 nmol = 1 140 µg C (Tsiafouli et al., 2015). The fungal biomass was the sum biomass of saprophytic fungal and AMF.

Nematodes were extracted from a 50 g soil sample (fresh weight) using a modified cotton-wool filter method (Liang et al., 2009). At least 100 nematode specimens from each sample were selected randomly and identified to genus level using an Olympus BX51 microscope (OLYMPUS, Tokyo, Japan) according to Bongers (1994). Following identification, the nematode length and maximum body diameter were determined with an ocular micrometer and the 145 nematode fresh biomass (µg) was estimated by multiplying the nematode length (µm) and maximum body diameter (µm) following Andárssy (1956). Nematodes were assigned into four trophic groups: bacterivores, fungivores, plant-parasites and omnivores-predators (Ferris, 2010).

Microarthropods were extracted from 200 mL fresh soil using modified high-gradient Tullgren funnels (Crossley and Blair, 1991) for 120 h at room temperature. Individuals were collected and stored in vials containing 95% ethanol for



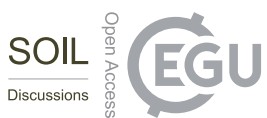

identification. Mites and collembolans were identified to species or morphospecies level according to Christiansen and Bellinger (1980-1981), Balogh and Balogh (1992), Bellinger et al. (2019), Pomorski (1998) and Niedbala (2002). Soil microarthropods were allocated into four different functional groups: fungivorous (oribatid) mites, predaceous mites, fungivorous collembolans and predaceous collembolans. Individual body length and width were measured to estimate the fresh biomass based on the conversion factor given by Berthet (1971) for mites and Edwards (1967) for collembolans.

## 2.4 Mineralization N of soil biota

The N mineralization of soil biota was estimated using the production-ecological model according to Didden et al. (1994) and Rashid et al. (2014) based on the C mineralization. The data for the biomass of bacteria and fungi, the fresh biomass of nematodes, mites and collembolans, the efficiency of assimilation ($A_e$) and production ($P_e$), and the C:N ratio of body and food were used for calculation of the mineralized N. The calculation was first conducted for each taxa using the number of individuals in the taxa, and then summed to get the values for their corresponding trophic group. Taking into account the changes in abundance of soil biota over time, the amount of N mineralization of different functional groups during the soybean growing season was estimated by summing the monthly N mineralization.

The values of $A_e$, $P_e$ and the body C:N ratio of soil biota were obtained from de Ruiter et al. (1993) and Didden et al. (1994) and are listed in Table S2. The food C:N ratios of bacteria, fungi and different functional groups of nematodes, mites and collembolans were calculated based on their food preferences and are given in Table S3. The food C:N ratios of bacteria and fungi were assumed to be equal to the C:N ratio of the detritus and roots, because bacteria and fungi are recognized as the primary decomposers (de Ruiter et al., 1993).

The respiration of bacteria and fungi at the field condition was calculated according to Rashid et al. (2014) as:

$$R_{BF} = Q_{10}^{(ST-T_0)/10} \times C_R \times B \times 30 \qquad (2)$$

Where, $R_{BF}$ is the respiration rate of bacteria or fungi (kg C ha$^{-1}$ month$^{-1}$); $Q_{10}$ is assumed to be 2.2 for both bacteria and fungi (Goulden et al., 1996); ST is the monthly mean soil temperature (°C); $C_R$ is the respiration rate constant which is 0.27 and 0.29 (kg C respiration per day per kg biomass) for bacteria and fungi, respectively, at $T_0$ (25 °C) (Stamatiadis et al., 1990); B is the biomass of bacteria or fungi (kg C ha$^{-1}$); 30 is the conversion from day to month.

The respiration of nematodes, mites and collembolans under field conditions was estimated by the conversion of oxygen consumption rate and calculated as:



$$R_t = O_t \times Q_{10}^{(ST-T_0)/10} \times TN_t \times 0.43 \times 10^{-3} \times 7.2 \qquad (3)$$

Where subscript t indicates the specific taxon of nematodes, mites or collembolans; $R_t$ is the C respiration rate of the

tth taxonomic group (kg C ha$^{-1}$ month$^{-1}$); $O_t$ is oxygen consumption rate of the tth taxonomic group (mm$^3$ $O_2$ individual

h$^{-1}$) at temperature $T_0$ (°C; Table S1); $Q_{10}$ is assumed to be 3.0, 3.0 and 2.5 for nematodes, mites and collembolans,

respectively (Persson et al., 1980); ST is the monthly mean soil temperature (°C); $TN_t$ is the total number of

individuals of the tth taxonomic group (individuals m$^{-2}$); $0.43 \times 10^{-3}$ is the conversion from mm$^3$ $O_2$ to mg C-$CO_2$ at

20 °C and standard pressure (100 kPa) using a respiration quotient ($CO_2$ expired / $O_2$ consumed = 0.8); 7.2 is the

conversion factor for up-scaling mg C m$^{-2}$ h$^{-1}$ to kg C ha$^{-1}$ month$^{-1}$. $O_t$ was calculated on the basis of the fresh body

mass (W) of the tth taxonomic group: $O_t = a \times W_t^b$ (Paplinska et al., 1972; Persson et al., 1980). The parameters a and

b of the tth taxonomic group are constants and are given in Table S2.

The C respiration was used to calculate C consumption ($C_0$), assimilation (A), production (P) and defecation (F) of a

given taxonomic group of soil biota (kg C ha$^{-1}$ month$^{-1}$) according to Persson et al. (1980) as:

$$C_0 = A / A_e \qquad (4)$$
$$A = R_t / (1 - P_e) \qquad (5)$$
$$P = A \times P_e \qquad (6)$$
$$F = C_0 \times (1 / A_e - 1) \qquad (7)$$

Where $A_e$ and $P_e$ are the production and assimilation efficiencies of tth taxonomic group, respectively; $R_t$ is the C

respiration rate of the tth taxonomic group (kg C ha$^{-1}$ month$^{-1}$) (Table S2).

The N mineralization of a given taxonomic group of soil biota ($N_{min}$; kg N ha$^{-1}$ month$^{-1}$) was calculated according to

Persson et al. (1983) and Rashid et al. (2014) as:

$$N_{min} = C_0 / C:N_{food} - F / (1.33 \times C:N_{food}) - P / C:N_{body} \qquad (8)$$

Where $C_0$, F and P are the C consumption, defecation and production of a given taxonomic group of soil biota (kg C

ha$^{-1}$ month$^{-1}$), respectively; $C:N_{food}$ and $C:N_{body}$ are the C:N ratios of food and a given taxonomic group of soil biota,

respectively; 1.33 is the conversion factor from food consumed C:N ratio to fecal C:N ratio (Persson et al., 1983).

**2.5 Energy pathways**

Energy pathways classified according to energy flowing into the trophic pyramids were used to describe the interaction

between plant and herbivorous fauna (i.e. root pathway), the interaction between microflora and microfauna (i.e.

microbial pathways) and the interaction between soil fauna and their predator (i.e.predator-prey pathway).



Microbial pathway was further divided into bacterial and fungal pathways due to the different nutrient turnover rates. Therefore, four pathways were categorized: (1) the root pathway, which represents the energy exchange between plant roots and their grazers (plant-parasitic nematodes); (2) the bacterial pathway, which represents the energy exchange between bacteria and bacterivorous nematodes; (3) the fungal pathway, which represents the energy exchange between fungi and their grazers (fungivorous nematodes, fungivorous mites and fungivorous collembolans); and (4) the predator-prey pathway, which represents the energy exchange between nematode, mite and Collembola prey (plant-parasitic, bacterivorus and fungivorous nematodes and fungivorous mites and collembolans) and their predators (omnivorous-predaceous nematodes, predaceous mites and predaceous collembolans) (de Vries et al., 2012; Holtkamp et al., 2011; Zhang et al., 2015).

Because the magnitude of biomass is different among functional groups (Table S4), the contribution of each energy pathway to soil mineralization N was standardized to avoid the dependence on its biomass (Holtkamp et al., 2008). The contribution of each pathway was the sum of its corresponding trophic group, which were standardized by dividing the overall seasonal mean of the contribution of that trophic group over all tillage treatments.

**2.6 Statistical analyses**

Response ratios of soybean yield, SNM, the mineral N delivered by soil biota and the contribution of mineralization N of soil biota to SNM were calculated to indicate the difference between conservation tillage and conventional tillage practices. A response ratio > 0 indicates a positive response to conservation tillage (i.e. RT and NT) relative to CT; conversely, a negative response ratio indicates a negative response to conservation tillage relative to CT. The tillage effects on these variables were tested using one-way or two-way analysis of variance (ANOVA) according to the number of fixed factors (i.e. tillage, soil depth and tillage × depth interaction).

Stepwise multiple linear regression (MLR) was used to identify and quantify the relationships between the crop yield and the N mineralization of each energy pathway at each soil depth. The stepping criteria employed for entry and removal were based on the significance level of the P-value and was set at 0.05. Before MLR, all parameters were min-max normalized to accurately preserve all relationships of data value and prevent potential bias from the domination of large numeric ranges over those with small numeric ranges. Min-max normalization subtracted the minimum value of an attribute from each value of the attribute and then divided the difference by the range of the attribute. The normalized value lay in the range [0, 1] (Jayalakshmi and Santhakumaran, 2011).

Stepwise MLR constructs a multivariate model for the dependent variable (Y) based on a few deliberately selected

explanatory variables. The best equation is selected on the basis of the highest coefficient of determination ($r^2$). The equation takes the following form: $Y = b_0 + b_1X_1 + b_2X_2 + ... + b_nX_n$ , where Y is the dependent variable (i.e. the crop yield); $X_1$, $X_2$, …, $X_n$ are the independent variables (i.e. the N mineralization of each energy pathway); $b_0$ is the constant, where the regression line intercepts the Y axis; $b_i$ ($1 \leq i \leq n$) is the standard regression coefficient and indicates the relative contribution of $X_i$ to the changes of Y. All statistical analyses were performed using the R software (R 3.4.0, R Development Core Team 2017) using the car package for ANOVAs and the stats package for MLR analyses.

## 3 Results

### 3.1 Soil mineralization N and soybean yield

The amount of soil mineralization N at 0–5 cm was higher ($P < 0.05$) in RT and NT than in CT during the soybean growing season, while the opposite trend was observed at 5–15 cm with a decrease of 25% in NT (Fig. 1). There was no statistical significance for soybean yield (Fig. S1); however, the yield of RT and NT increased by 6.6% and 26.5%, respectively, in comparison with CT (Fig. 1).

### 3.2 Mineralization N of soil biota

The amount of mineralization N of soil biota was highly dependent on the magnitude of their biomass and generally decreased with an increase of trophic level in all tillage practices at both soil depths (Table S4 and S5). However, when considering tillage effect, higher trophic groups responded to conservation tillage more obviously than lower trophic groups (Fig. 2a). For example, RT and NT increased the amount of mineralization N of bacterivorous nematodes, predaceous mites and predaceous collembolans by 142%, 134% and 71% respectively at 0–5 cm and of omnivorous-predaceous nematodes by 40% at 5–15 cm, while they increased the amount of mineralization N of bacteria and fungi by only 21% and 36% respectively at 0–5 cm and 24% and 28% at 5–15 cm (Fig. 2a).

### 3.3 Relationship between soil biota and soil mineralization N

The contribution of most trophic groups to soil mineralization N responded positively to conservation tillage practices at both soil depths (Fig. 2b). Among all trophic groups, the contributions of bacterivorous nematodes and omnivorous-predaceous nematodes to soil mineralization N were the greatest in NT at 5–15 cm, respectively increasing by 147% and 143% (Fig. 2b).





Predator-prey and fungal pathways were the main contributors to soil mineralization N in all tillage practices at both soil depths (Fig. S2). However, with the exception of the fungal pathway, the contribution of other energy pathways varied with tillage at both soil depths. Among these tillage affected pathways, root, bacterial and predator-prey pathways both in 0–5 cm and in 5–15 cm had the highest positive response ratio to NT, while the bacterial pathway in

0–5 cm had the highest response ratio to RT, increasing by 65% (Fig. 3).

### 3.4 Relationship between soil biota and soybean yield

At 0–5 cm, the stepwise regression model included the root, fungal and predator-prey pathways and accounted for the 89% variation of the soybean yield (Table 1). The relative contributions of these pathways to the soybean yield

decreased in the order of root pathway (0.487) > fungal pathway (0.389) > predator-prey pathway (0.318). At 5–15 cm, the model only included the bacterial pathway and explained 55% variation of soybean yield (Table 1).

## 4 Discussion

### 4.1 Contribution of soil biota to soil mineralization N

The data have shown that most trophic groups release more mineral N in RT and NT soils than in CT in the plough layer during the soybean growing season after 15-yr continuous application of conservation tillage. However, these differences between CT and RT and NT were more pronounced for higher trophic groups than for lower trophic ones. Our result is inconsistent with the general viewpoint that the trophic groups with a small population and a long distance from the basal resources have a less important role in regulating nutrient mineralization than those trophic

groups with a large population and a short distance (Cole et al., 2002; Deruiter et al., 1993; Hunt et al., 1987). However, our result confirms the recent reports of Holtkamp et al. (2011) and Rashid et al. (2014) who estimated the contribution of higher trophic groups to soil N mineralization and suggested that a considerable contribution was originated by the higher trophic groups through their individual metabolism and their effect on their prey.

The changes in trophic groups were also transferred into the energy pathways, where the contribution of bacterial and

predator-prey pathways to soil N mineralization in the whole plough layer increased significantly in RT and NT. This may primarily result from the higher connectance of these two pathways in RT and NT than in CT (Zhang et al., 2015). The enhanced connectance among trophic groups promotes higher nutrient turnover rate because it increases the energy flow from lower trophic groups to the higher ones (Morriën et al., 2017). Therefore, consecutive interactions among the main components (bacteria, bacterivorous nematodes and their predators) of bacterial and predator-prey



pathways could exist in RT and NT soils, which facilitates N mineralization delivered by soil organisms.

## 4.2 Contribution of soil biota to plant yield

The multiple regression models showed that there was a significant correlation between the mineral N delivered by different energy pathways and soybean yield, supporting the hypothesis that the increase in N mineralization by soil

biota is responsible for the higher soybean yield in RT and NT. This is consistent with the reports of Bender et al. (2015) and Evans et al. (2011) that were also conducted in field conditions and suggests that farming practices favoring a rich and abundant soil biota can improve crop yield by increasing N availability to plants. Nevertheless, our data further suggest that the capacity of energy pathways in mediating N supply to plants was different, as indicated by standard regression coefficient.

Root pathway at 0–5 cm was the greatest contributor to the high soybean yield in RT and NT since it increases N availability to plant. This seems contrary to the popular opinion of root pathway, namely energy exchange between plant root and plant-parasitic nematodes, has a considerable threat to the crop yield potential (Nicol et al., 2011). Actually, nematodes that do substantial damage to plant growth are the obligatory root feeders such as Heterodera glycines (soybean cyst nematode). However, in this study, most genera classified as plant-parasitic nematodes, such as

the Boleodorus, Tylenchus and the Basiria (data not shown), are the facultative root feeders, which feed on both the root epidermal cells and the fungal hyphae (Yeates et al., 1993). Sohlenius et al. (2011) reported that resource enrichment is the driving factor in the feeding preference of this group. Since the hyphal branches develop better with less tillage disturbance under RT and NT than under CT (van Capelle et al., 2012), fungal hyphae may act as an alternative food source and diminish the adverse effect of facultative feeders on plant growth.

In contrast to the detrimental effects on the crop, plant-parasitic nematodes can sometimes stimulate host growth and improve crop yields in a low grazing pressure scenario (Bonkowski, 2009). This may due to the fact that the plant-parasitic feeders increase the allocation of carbon to rhizosphere microorganisms, which in turn supply nutrients to plant (Bonkowski, 2009; Bonkowski et al., 2000). Additionally, considering the significant contribution of predator-prey pathway to the plant yield at the same soil depth (0–5 cm), the excessive damage of plant-parasitic

nematodes to soybean growth might be inhibited by the the high trophic level of predators, i.e. predaceous mites. Our results were supported by Verschoor (2002) who calculated the N budget of plant-parasitic nematodes in ryegrass and reported that the N mineralization of plant-parasitic nematodes accounts for 4%–8% of the total standing biomass N and increased growth from this additional N offsets any plant damage imposed by the plant parasitic nematodes.



Overall, we propose that the beneficial effect of root pathway on plant growth in RT and NT soils is a synergy of fungal and predator-prey pathways.

Fungal pathway as well as bacterial pathway indicating the interaction between microflora and microfauna have been considered to play a critical role in influencing soil N mineralization and N uptake by the plant (Bardgett and Chan, 1999; Bonkowski et al., 2000; van Groenigen et al., 2015). This was revealed in RT and NT soils where the N mineralized by fungal pathway at 0–5 cm and by bacterial pathway at 5–15 cm strongly contributed to the soybean yield. However, in contrast to bacterial pathway which was the dominant contributor at 5–15 cm, fungal pathway coupled with the root and predator-prey pathways were the driving contributors at 0–5 cm to the increase in soybean yield in RT and NT. Similar results were also reported by Bender and van der Heijden (2015), Cole et al. (2002), Gange (2000) and Holtkamp et al. (2011), who suggested that the beneficial effect of fungal pathway on N mineralization is limited and only appeared when they interacted with other trophic groups. This may be partially explained by the 'slow' interior character of fungal pathway and that the components within the fungal pathway have longer generation times and process material and energy at slower rates in comparison with the bacterial pathway (Moore et al., 2005). This leads to an obscure relationship between soil N mineralization and fungal pathway (Rooney et al., 2006). Furthermore, a detrimental effect on the N transfer from arbuscular mycorrhizal fungi to plant might happen by over grazing by the fungal-feeding grazers alone (Gange, 2000). The presence of other trophic groups may mitigate this detrimental effect by forming different associations, i.e. competition, collaboration and predation, with the components of fungal pathway.

Predator-prey pathway, which indicates the interaction among soil fauna, was the third contributor at 0–5 cm soil depth to enhancing soybean yield of RT and NT. This may due to the increase in the abundance of predators under RT and NT soils. Predator-prey pathway may affect the N mineralization directly through the excretion of extra N after prey consumption and indirectly through the changes in the prey community by the top-down effect (Moore et al., 2003; Moore et al., 2005; Holtkamp et al., 2011). Predator-prey pathway was working together with plant and fungal pathways in explain the variation of soybean yield at 0–5 cm, therefore we propose that the top-down effect may the driving factor in increasing the N availability to plants since predators can control the risks that some specific species of plant and fungal pathways will exert a negative influence on plant growth.

## 4.3 Multi-cooperation of soil biota for agricultural sustainability

Our results highlight that the synergy among different pathways is the key to yielding more grain under RT and NT



than under CT since it can effectively couple N mineralization and plant uptake. This effect was particularly revealed in the upper soil layer (0–5 cm) of RT and NT soils in which soil biota interacted through plant, fungal and predator-prey pathways in the horizontal perspective of soil profile to increasing N availability to plants (Table 1 and Fig. 4).

From the vertical perspective of soil profile, the pathway that regulates the N availability to plants changed from the bacterial pathway that was dominant in the whole plough layer under CT to the fungal and bacterial pathways that dominated at different soil layers under RT and NT (Fig. 4). Empirical studies have shown that the co-existence of fungal and bacterial pathways is the key to agricultural sustainability, because large shifts in nutrient cycling toward either the fungal pathway or the bacterial pathway may result in unstable ecosystem service (Moore et al., 2005; Rooney and McCann, 2012). Our result suggests that there is a balance between fungal and bacterial pathways in the profile of RT and NT soils, which may play a synergistic role in N mineralization in providing sufficient mineralized N for the vertically distributed root absorption sites. This may partially explain why the deficit in the soil mineralization N amount at 5–15 cm of RT and NT during the growing season did not result in compromise of soybean yield (Fig. 1).

Top predators of predator-prey pathway, which couple the energy flow from fungal and bacterial pathways, are the balance regulators which prevent ecosystems from being dominated almost completely by a specific energy pathway (Moore et al., 2005). The results of multiple regression models did not emphasize the coupling effect between predator-prey pathway and bacterial pathway on soybean yield at 5–15 cm. However, the significant increase in N mineralization of predaceous mites at 0–5 cm and omnivorous-predatory nematodes at 5–15 cm under RT and NT imply that predator-prey pathway may act as a coupler of fungal pathway in 0–5 cm and bacterial pathway in 5–15 cm, because the high density of predators may intensify the vertical migration for preying. Furthermore, this implication was also supported by the significant relationship (Fig. S3) between microbial (i.e. fungal, bacterial and fungal + bacterial) pathways and predator-prey pathway in N mineralization throughout the plough layer (0–15 cm). Our result was supported by the report of Doblas-Mirand et al. (2009) that the energy flux in different soil layers can be connected through the vertical movement of soil fauna. Therefore, similar to the cooperation of soil biota in the horizontal plane, soil biota in the vertical plane might be also cooperate through different energy pathways to maintain a stable supply of inorganic N to plant under RT and NT (Fig. 4).

**5 Conclusion**

It is undeniable that in addition to the effect of soil biota on N mineralization, soil N mineralization is also related to
the fixation of N through soybean root nodules, and the loss of mineral N via runoff, leaching, and denitrification to N$_2$O. This study was focused on addressing how soil biota act as the mineral N engineer in maintaining crop productivity under contrasting tillage practices. Our findings have shown that soil biota exert a positive effect on

coupling soil N mineralization and soybean productivity throughout the whole plough layer after long-term application of conservation tillage. Although soil trophic groups involved in coupling N mineralization and soybean yield varied with soil depths, they cooperated through different energy pathways to connect the N flow horizontally and vertically. This has resulted in a sustainable flow of mineral N from soil biota to plant in the plough layer, which is the key for maintaining a high grain yield in conservation tillage.

**Author contribution**

S.X.Z, H.T.W and A.Z.L designed research; S.X.Z, S.Y.C and L.C performed research; W.J.L and W.D.H guided species classification; S.X.Z analyzed data; and S.X.Z, N.B.M, H.T.W and A.Z.L wrote this paper.

**Competing interests**

The authors declare that they have no known competing financial interests or personal relationships that could have appeared to influence the work reported in this paper.

**Acknowledgments**

We would like to thank Dr. Su Zhang (Changchun Institute of Applied Chemistry, CAS) for graphic edit. This research was supported by the National Natural Science Foundation of China (No. 41401272 and 41430857), the Foundation of Excellent Young Talents in Northeast Institute of Geography and Agroecology, Chinese Academy of Sciences (DLSYQ15001), the Jilin Province Science and Technology Development Plan Project (20190201116JC), and the Key Research Program of Frontier Sciences of Chinese Academy of Sciences (QYZDB-SSW-DQC035).

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



**Table 1** Stepwise regression equations of soybean yield against N mineralization of bacterial, fungal and predator-prey pathways of 0–5 and 5–15 cm depths. Both soybean yield and N mineralization pathways were min-max normalized and are dimensionless.

| Soil depth (cm) | Regression equation | $R^2$ | P value |
|---|---|---|---|
| 0–5 | y = -0.18 + (0.487 × root pathway) + (0.389 × fungal pathway) + (0.318 × predator-prey pathway) | 0.89 | < 0.001 |
| 5–15 | y = -0.093 + (0.770 × bacterial pathway) | 0.55 | 0.003 |






**Figure Legends**

**Fig. 1** The response ratio (mean ± standard error) of soil N mineralized (SNM) during the growing season and soybean yield between conservation tillage (RT, ridge tillage; NT, no tillage) and conventional tillage (CT) practices. *, significant differences between tillage practices at the P < 0.05 level.

**Fig. 2** The response ratio (mean ± standard error) of mineralization N delivered by soil biota (a) and of the contribution of soil biota to soil mineralization N (b) between conservation tillage (RT, ridge tillage; NT, no tillage) and conventional tillage (CT) practices at different soil depths during the soybean growing season . Total, total soil trophic groups; Ba, bacteria; Fu, fungi; NBF, bacterivorous nematodes; NFF, fungivorous nematodes; NPP, plant-parasitic nematodes; NOP, omnivorous-predaceous nematodes; MFF, fungivorous mites; MPR, predaceous mites; CFF, fungivorous collembolans; CPR, predaceous collembolans; '*', '**' and '***' indicate significant differences between tillage practices at the P < 0.05, < 0.01 and < 0.001 level, respectively.

**Fig. 3** The response ratio (mean ± standard error) of the contribution of different energy pathways to soil mineralization N between conservation tillage (RT, ridge tillage; NT, no tillage) and conventional tillage (CT) practices at different soil depths during the soybean growing season. '*' indicates significant differences between tillage practices at the P < 0.05 level. RP, root pathway; FuP, fungal pathway; BaP, bacterial pathway; PpP, predator-prey pathway.

**Fig. 4** Flow diagram illustrating how soil biota interact (indicated by the black arrows; solid black arrows indicate the N mineralization depends on the size of basal resources; dashed black arrows indicate the control in N mineralization through the relationships of predation, competition and collaboration between trophic groups) through different energy pathways in the plough layer to couple N mineralization and plant growth under contrasting tillage practices (A, conventional tillage; B, conservation tillage). Soil biota exhibited synergy through plant, fungal and predator-prey pathways at the horizontal level (0–5 cm), and through the energy balance between fungal pathway and bacterial pathway at the vertical level (0–15 cm) to increasing N availability to plant. Green arrow indicates the energy flow originated from resources (i, root residues and excretions; ii, crop residues) into soil biota. Blue arrow indicates the provision of mineralization N to plant through soil biotic activity. Arrow thickness indicates the magnitude of energy and N flow. Red –'X' indicates that three are no significant synergistic effects among soil biota.



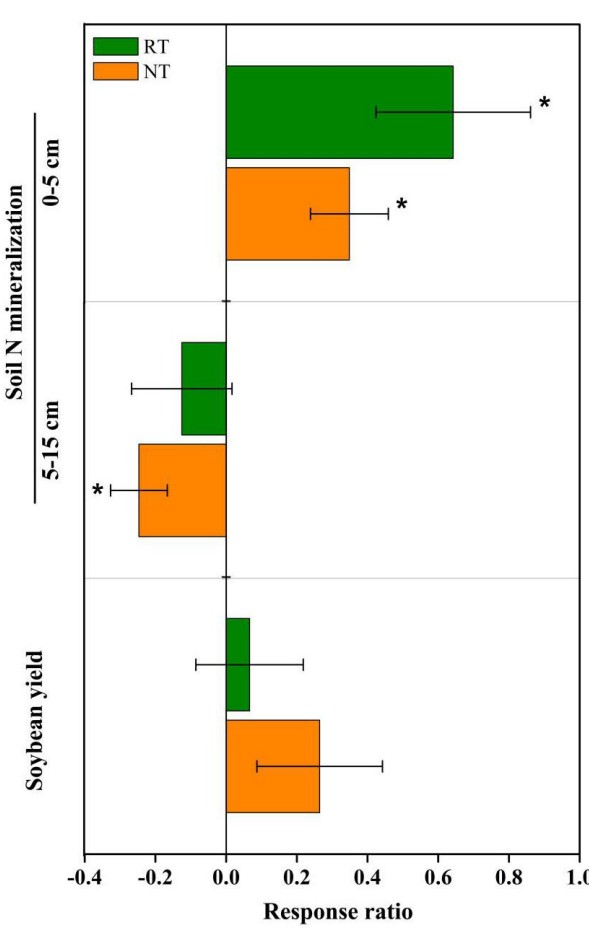

**Fig. 1** (Shixiu Zhang)





**Fig. 2** (Shixiu Zhang)







**Fig. 3** (Shixiu Zhang)





**Fig. 4** (Shixiu Zhang)