# Peer review of "application of conservation tillage system in a black soil of"

_SOIL, 2020_

## Referee Comment (RC1) · Anonymous Referee #1 · 10 Mar 2020

Manuscript Title: Multi-cooperation of soil biota in the plough layer is the key for conservation tillage to improve N availability and crop yield Manuscript Number: Soil-2020-2 The manuscript examines the influence of soil biota on coupling N mineralization with soybean crop yield under conventional and conservation tillage regimes. Many studies already existed in the literature, which investigated the influence of soil biota to soil N mineralization using foodweb modeling (De Ruiter et al, 1993a, b; De Ruiter et al, 1994; Hunt et al. 1987; Schon et al. 2013) approach in different ecosystems (grassland or arable). In addition, other studies are also available in the literature who used production ecological calculation to study contribution of soil biota to N mineralization (Persson et al. 1980; Persson 1983; Didden et al. 1994, de Goede et al. 2003, Van Vliet et al.

2007; Rashid et al. 2014; Deru et al. 2019). Most of these studies are carried out in the Netherlands and/other European countries however the current study investigated N mineralization and coupling that with crop yield under different tillage regimes in China. Therefore, it would be a good addition to the scientific knowledge in this field. However, there are many flaws in the parameters used to calculate N mineralization or methodology that need to be considered carefully to make this manuscript acceptable for the publication. Material and Methods What was the motive behind choosing 0-5 and 5-15 cm soil layer for soil biota sampling and N mineralization when the plow layer for conventional tillage was 20 cm? For the latter case, tillage operation mixed the soil layer of 0-20 cm. Why bulk density was recorded at 5 cm and 10 cm and not 0-5 and 5-15 cm soil depth? The difference in bulk density might affect the soil N mineralization. Line 95, Zhang et al. (2019) used 40 kg N ha-1 in the soybean field. Moreover, there might be atmospheric N deposition. Therefore, all or part of the N from the applied fertilizer and/or atmospheric N deposition can be taken up by soybean and help to increase the yield of soybean, how this effect of N fertilization on crop N uptake/yield was separated from N contribution by a different trophic group of soil organisms? Please explain why N fertilizer (40 kg N ha-1) plot wAS considered as a suitable reference to estimate background crop yield/N response. Line 107, please add the soil depth at which temperature was recorded. Line 119, Why the mineral N before incubation was measured and not after one week in the potential N mineralization method, ideally mineral N can be subtracted after 1 week of incubation. Since this time frame is used to enumerate the biota activity at optimal temperature and moisture content. Therefore, N mineralized during this time would be low and if this deleterious effects would not be adjusted then this effect may lead to underestimation of N mineral from the soil (Bloem et al. 1994). Line 148, please add the soil layer in cm where microarthropods were extracted? If the soil sample were collected from 15 cm soil depth, from the current unit it is not clear whether these organisms were extracted from 0-15 cm soil layer or 0-7.5 cm. How their contribution would be related to actual soil N mineralization from 0-5 and 5-15 cm? Although biota biomass from table 4 indicates the presence of these organisms in 0-5 and 5-15 cm, this should be explained in the methodology, in which depth actually the organisms were extracted. Earthworms were not present in this system or these organisms were not sampled from this experiment. Most of the published studies indicated their significant contribution after bacteria and fungi to N mineralization, it would be good to include their contribution in such systems. Table S3, Why actual C:N ratio of the root of the soybean crop studied was not used. The currently used C:N ratio of soybean root is much less than the actual C:N ratio of the soybean, see for example (Kushwah et al. 2014; Redin et al. 2018). Such lower C:N ratio used in the calculation could lead to high N mineralization and hence overestimation of N mineralization in this category. Table S4, values of biotic biomass were expressed in mg C m-2, but I could not find the reference of Berg et al. (1998) to confirm the average C content of 48% dry biomass used for microarthropods. For the nematodes biomass C, Ferris (2010) also adjusted this 0.1 C factor by using the formula, Pt= 0.1 Wt/mt, where Pt, Wt and mt are the C used in production, the body weight, and the cp class of taxon t, respectively. However, these factors may also influence the C biomass which may lead to over/underestimation of the biomass and therefore N mineralization by this group of soil biota. The table S5, the contribution of N mineralization by a different group of soil organisms, these result are the main result according to the objective of the study, therefore can be moved to the main manuscript. Line 163, please add the data about the number of taxa or abundance of soil organisms (nematodes and microarthropods in supplementary information or main text). Line 250, No difference in soybean yield among different treatments might be linked with the applied N fertilizer dose and/or atmospheric N deposition. Moreover, can you please explain why the difference in soil N mineralization among different treatments would not result in yield increment of soybean among different treatments? It seems yield was tended to be higher but did not differ significantly among treatments. Would the presentation of crop N uptake rather than crop yield explain the difference? Fig. S1a, the P-values presented in the figure indicate that tillage, depth, and their interaction were significant, please use multiple comparisons to differentiate the effects, if done already please add letters on the bar to

differentiate the effect of the treatments within or between the two depths. These are the main result, therefore, I suggest presenting them in the main manuscript rather than supplementary information. Line 251, presenting the biomass or abundance data in the main manuscript would add more value, therefore I would suggest adding this data in the manuscript. Line 263, indicate that bacterivorous nematodes and omnivorous-predaceous nematodes contributed highest to N mineralization that was not the case in Table S5. Can you please discuss this difference in detail in the discussion section? Or I could not understand from the current formulation what do you mean? Fig. 1 The response ratio of soil N mineralized during the growing season and crop yield was calculated, is that a fair comparison. Is it not better to use the response ratio of soil N mineralization and crop N uptake? To check for the accuracy of modeling: did the temporal variation in calculated N-mineralization rates correspond with the temporal variation in measured N-mineralization rates (potential N mineralization)? I could not see this in the manuscript. The main aim of the manuscript is to examine the influence of soil biota on coupling N mineralization with soybean yield therefore the current fig. 1 did not meet the objective. Hence, I would suggest to also include the response ratio of soil N calculated based on the modeling and soybean yield. Fig. 2, what is the difference between mineralization N delivered by soil biota and of the contribution of soil biota to soil mineralization N? Please clarify it. Discussion Line 285, In the case of Holtkamp et al. (2011) bacteria and fungi contributed about 77% of the total N mineralized which is in line with Rashid et al. (2014), who estimated that the aforementioned biota contributed to the 60% of the soil N mineralized. So, bacteria and fungi but not the higher trophic groups were responsible for most of the soil N mineralization in their systems. Even in your system Table S5, the contribution of fungi is the highest followed by bacteria and there is an insignificant contribution to N mineralization is coming from nematodes and microarthropods. What do you mean by the higher trophic group here? Lines 328-330, why fungal pathways were dominated in the soil layer 0-5 and bacterial pathways in the layer 5-15 cm in RT and NT tillage? Can you please mechanistically explain how these pathways contributed to soybean yield? In lines 335-341, I expected

the discussion on why the fungal pathways were dominated contributors of soybean yield in 0-5 cm and bacterial pathways in 5-15 cm soil layer? Can you please discuss further how and why these pathways were dominated in these layer under RT and NT tillage operations. The manuscript uses modeling to estimate various fluxes of N in the soybean. In the model, a lot of parameters were taken from literature rather than from measurements in the actual sites. What the authors fail to discuss (and to mention), is that there is a degree of uncertainty associated with any model. Each estimate based on modelling equations comes with the error range. Depending on the model and the parameter in question, this error range can be small or large. Therefore, a sensitivity analysis should be carried out. Moreover, it needs to be mentioned, if any conclusions are to be drawn based on model-derived numbers. A model estimate for any parameter should never be presented as a single number without an error range. I encourage the authors to reflect this in the Discussion and Conclusion. Please provide the error range for the values you estimate based on models, and please adjust your Discussion of differences in soil N fluxes, and your Conclusions, to reflect the uncertainties associated with modeling.

References De Goede, R.G.M., Brussaard, L., Akkermans, A.D.L., 2003. On-farm impact of cattle slurry manure management on biological soil quality. NJAS - Wagen. J. Life Sci. 51, 103–133. De Ruiter, P.C., Moore, J.C., Zwart, K.B., Bouwman, L.A., Hassink, J., Bloem, J., Vos, J.A.D., Marinissen, J.C.Y., Didden, W.A.M., Lebrink, G., Brussaard, L., 1993a. Simulation of nitrogen mineralization in the below-ground food webs of two winter wheat fields. J. Appl. Ecol. 30, 95–106. De Ruiter, P.C., Veen, J.A., Moore, J.C., Brussaard, L., Hunt, H.W., 1993b. Calculation of nitrogen mineralization in soil food webs. Plant Soil 157, 263–273. De Ruiter, P.C., Neutel, A.M., Moore, J.C., 1994. Modelling food webs and nutrient cycling in agro-ecosystems. Trends Ecol. Evol. 9, 378–383. Deru, J.G., Bloem, J., de Goede, R., Hoekstra, N., Keidel, H., Kloen, H., Nierop, A., Rutgers, M., Schouten, T., van den Akker, J. and Brussaard, L., 2019. Predicting soil N supply and yield parameters in peat grasslands. Applied soil ecology, 134, pp.77-84. Didden, W.A.M., Marinissen, J.C.Y., Vreeken-Buijs, M.J., Burgers,

S.L.G.E., de Fluiter, R., Geurs, M., Brussaard, L.,1994. Soil meso- and macro-fauna in two agricultural systems: factors affecting population dynamics and evaluation of their role in carbon and nitrogen dynamics. Agric. Ecosyst. Environ. 51, 171–186. Ferris, H.: Form and function: Metabolic footprints of nematodes in the soil food web, European Journal of Soil Biology, 46, 97–104, doi: 10.1016/j.ejsobi.2010.01.003, 2010 Hunt, H.W., Coleman, D.C., Ingham, E.R., Ingham, R.E., Elliott, E.T., Moore, J.C., Rose, S. L., Reid, C.P.P., Morley, C.R., 1987. The detrital food web in a shortgrass prairie. Biol. Fertil. Soils 3, 57–68. Kushwah, S.K., Dotaniya, M.L., Upadhyay, A.K. et al. Assessing Carbon and Nitrogen Partition in Kharif Crops for Their Carbon Sequestration Potential. Natl. Acad. Sci. Lett. Persson, T., Bååth, E., Clarholm, M., Lundkvist, H., Söderström, B.E., Sohlenius, B., 1980. Trophic structure, biomass dynamics and carbon metabolism of soil organisms in a Scots Pine Forest. Ecol. Bull. 419–459. Persson, H., 1983. Influence of soil animals on nitrogen mineralization in a northern Scots pine forest. In: Lebrun, P., Andre, H.M., de Medts, C., Gregoire-Wibo, Wanthy, G (Eds.), New Trends in Biology. Dieu-Brichart, Louvain-la-Neuve, pp.117–126. Rashid, M.I., de Goede, R.G., Brussaard, L., Bloem, J. and Lantinga, E.A., 2014. Production-ecological modelling explains the difference between potential soil N mineralisation and actual herbage N uptake. Applied soil ecology, 84, pp.83-92. Redin, M et al. Root and Shoot Contribution to Carbon and Nitrogen Inputs in the Topsoil Layer in No-Tillage Crop Systems under Subtropical Conditions. Rev. Bras. Ciênc. Solo [online]. 2018, vol.42 Schon, N., Mackay, A., Hedley, M., Minor, M., 2012. The soil invertebrate contribution to nitrogen mineralisation differs between soils under organic and conventional dairy management. Biol. Fertil. Soils 48, 31–42. Van Vliet, P.C.J., van der Stelt, B., Rietberg, P.I., de Goede, R.G.M., 2007. Effects of organic matter content on earthworms and nitrogen mineralization in grassland soils. Eur. J. Soil Biol. 43, S222–S229. Zhang, Y., Li, X., Gregorich, E.G., McLaughlin, N.B., Zhang, X.P., Guo, F., Gao, Y. and Liang, A.Z.: Evaluating storage and pool size of soil organic carbon in degraded soils: Tillage effects when crop residue is returned, Soil & Tillage Research, 192, 215–221, doi: 10.1016/j.still.2019.05.013, 2019.
Please also note the supplement to this comment:
https://www.soil-discuss.net/soil-2020-2/soil-2020-2-RC1-supplement.pdf
* * *

---

## Referee Comment (RC2) · Anonymous Referee #2 · 6 Apr 2020

This manuscript needs significant improvements in English grammar to be understandable and publishable.

Title: needs re-working. "Multi-cooperation" isn't correct. Perhaps simply "interaction"? Affiliations: I think there should be a better translation for "Key Laboratory of Mollisols Agroecology". Even simply "Laboratory of Mollisol Agroecology" Ln 13: Please check English grammar. For example, "Conservation tillage systems may promote more complex and heterogeneous distributions of soil organisms relative to conventional tillage that may result in higher crop yield. However, the role of soil biota in N mineralization promoting plant growth remains limited." Some introduction or definition of "trophic

groups" and "energy pathways" is needed. Ln 27-31: Is the second to last sentence of the Abstract the main finding of the study? The last statement, on lines 30 and 31, is quite a broad generalization and is not overly useful. The second to last sentence here, lines 27 to 30 would seem to say that ploughed and non-ploughed systems are similar in terms of N supply to plants, is that what you mean? Clarification may be needed.

---

## Referee Comment (RC3) · Anonymous Referee #3 · 7 Apr 2020

The authors present data from a long term (14yr) tillage trail comparing conventional, ridge and no tillage and their effects on soil biota contributions to N mineralization and crop yield. While the paper highlights potentially interesting aspects of biological soil functioning, I am not convinced that the methods that have been applied are justified for the authors' goals.

Specific comments One key concern is that N mineralization is measured under laboratory conditions and then corrected to field conditions, via a solely temperature-dependent Q10 equation (L112-114). It is well known that the simple Q10 relationship does not hold under realistic soil conditions, since temperature is not the only limiting

factor. Soil moisture, substrate availability, etc also strongly co-determine the biogeo-chemical process rates in situ (see e.g. Davidson & Jansses 2006 Nature 440: 165-173 for SOM decomp). Therefore, I do not believe that the authors can capture realistic N mineralization rates in their field. I think this paper needs a thorough validation of this relationship.

Similarly, I am highly critical of the way the authors attribute N mineralization contributions from different soil biota groups. They use a series of equations from other authors to transform soil biota abundances into process rates (e.g. L170-L176, L177-188, L198-202). Mostly these steps seem to be based on Rashid et al 2014. These steps form the heart of their study. For instance, the conclusion that conservation tillage promotes N min (L21-23), hinges on these equations that all assume that more soil biota lead to more N min. The same goes for the relative contributions of soil biotic groups to total N mineralization (L25-27). The parameter estimates (e.g. Q10 of 3, L116) used come from different systems in other countries, while it is know that N cycling processes are highly heterogeneous in space and time. I am therefore sceptical that the same relations and the same parameter estimates will hold in the system studied by the authors. In fact even in the source paper, Rashid et al 2014, the ecological-production model is an improvement over the standard government rules, but still there is considerable error in the estimates (87-120% of observed N min rates) on the fields they studied. So I think the authors have to spend much more effort on convincing me and other readers that using these equations leads to valid inferences about this particular system. To be honest, as an empiricist, I think that to only realistic way to get to these questions is to use isotopic tracers in the field plots. However, what would help is if 1) we had realistic data on N min rates in the actual plots, and 2) the summed N contributions over the soil biota would have a strong predictive relationship with these independent field data. As it stands such a field validation is totally missing, which makes the study unconvincing.

Data were missing in some months for nematode data and linear interpolation was

used to fill these data gaps (L129). I find this a risky approach, especially since nematode population dynamics within season are non-linear, see e.g. the data in Rashid et al, but also other sources. I think the authors also need to show that their conclusions hold if the only work with the months where they have data on all soil groups.

The authors use the ratios of (calculated) mineral N delivery in the conservation tillage (ridge, and no tillage) to conventional tillage in their main figures. However, ratios are biased (e.g. Jasieński & Bazzaz 1999 Oikos 84: 321-326); a log(Treat/Control) has better statistical properties (Brinkman et al 2010 J Ecol 98: 1063–1073). Even better however would be if the main analyses and figures are directly based on the data from the three treatments directly, this approach would even give you a bit more statistical power. In that sense I find the supplementary figures to be much clearer.

In general, I find that the writing is a bit to colloquial in tone and imprecise in many places. See some examples below. Also I find that the presentation of the energy channels to be a bit overstated, there have been many findings of cross-feeding across these 'channels', and really I think we need to adopt a network view of the soil community and its links to biogeochemical processes.

Minor comments - L44: what do you mean with 'special species'? - L51: what are weak root infections - L55: what do you mean by capacity? Use of substrates? Process rates? - L60: I would not use the word conquer here, maybe mediate? - L61: adverse effects on what? - L66: rich in what sense - L68: what is stratified and in what way? - L80: based on M&M I believe its 14 years, not 15. - L83: what do you mean coupling? How will you quantify that coupling? - L85: it is a bit unclear what you mean by multiple spatial interactions in this hypothesis. How will you test this? - L94: how big were the plots? - L100: what was done with the maize residue?

---

## Editor Comment (EC1) · Elizabeth Bach (Editor) · 14 Apr 2020

Dear Zhang et al, Thank you for your patience, I have now received two complete peer reviews of your manuscript "Multi-cooperation of soil biota in the plough layer is the key for conservation tillage to improve N availability and crop yield" submitted to SOIL. The reviewers found the work a useful contribution to the scientific literature investigating soil biological contributions to N mineralization in agricultural systems. However, both reviewers raise concerns with how models from the literature were applied to this specific study. Applying models can offer insights and predictions, but it is important to understand and report the uncertainties that arise from inputting field and laboratory

data from one study into a model developed in another. A way to address this would be to conduct a sensitivity test, as suggested by reviewer 1. Additionally, caveats need to be incorporated throughout the results and discussion, especially to the conclusions, which both reviewers felt were overstating the underlying data.

Both reviewers mention ways the text could be improved for clarity. In some cases, there is confusion around methods, which may require some extensive rewriting. It is important to consider where grammatical changes can improve the text and where additional information is truly required. A third reviewer found the writing too confusing to do a full reviewer; however, the thorough review of the other two reviewers provides sufficient feedback to proceed with revision of the manuscript.

Please proceed with responding to the reviewer comments. After those responses are posted, I will proceed with a decision on the manuscript.

Thank you, Elizabeth Bach Topical Editor, SOIL

———————————————————

---

## Author Comment (AC1) · 26 May 2020

1.Material and Methods What was the motive behind choosing 0-5 and 5-15 cm soil layer for soil biota sampling and N mineralization when the plow layer for conventional tillage was 20 cm? For the latter case, tillage operation mixed the soil layer of 0-20 cm. Why bulk density was recorded at 5 cm and 10 cm and not 0-5 and 5-15 cm soil depth? The difference in bulk density might affect the soil N mineralization.

Soil stratification is a typical characteristic of conservation tillage, because there is a contrasting difference between top soil (usually means 0-5 cm) and the sub soil. Using either 5-15 cm or 5-20 cm to investigate the conservation tillage effect on the sub-soil

depth is very common in the literature (for example, 5-15 cm in the study of Gómez-Rey, et al., 2012; 5-20 cm in Haplern et al., 2010). Our previous study found that there was no significant difference between these two soil depths (5-15 cm and 5-20 cm) in soil C, N, bulk density, soil water content, and the other parameters of soil physicochemistry, but there was a slight difference in the abundance of soil collembolans and mites. Their abundance at the 20 cm was very low. So, on this basis, we think it is more reasonable to use 5-15 cm to investigate the role of soil organisms. We rewrote the description in the paper about how the soil bulk density was determined.

2.Line 95, Zhang et al. (2019) used 40 kg N ha-1 in the soybean field. Moreover, there might be atmospheric N deposition. Therefore, all or part of the N from the applied fertilizer and/or atmospheric N deposition can be taken up by soybean and help to increase the yield of soybean, how this effect of N fertilization on crop N uptake/yield was separated from N contribution by a different trophic group of soil organisms? Please explain why N fertilizer (40 kg N ha-1) plot wAS considered as a suitable reference to estimate background crop yield/N response.

We focused on investigating the difference of soil organisms among different tillage systems, not on the soil input N. Furthermore, the amount of soil input N as fertilizer is the same in all tillage systems; the amount of N fertilizer applied is typical for soybean crop grown by local farmers. For the deposition of atmospheric N, its contribution can be neglected even if it is not uniformly distributed in the atmosphere, because it is very small relative to the amount of nitrogen fertilizer; further, all plots in the experimental site would receive the same deposition from the atmosphere. Therefore, in this context, there would be no significant difference in the utilization of N in soybean of the same variety.

3.Line 107, please add the soil depth at which temperature was recorded.

Soil depth has been added.

4.Line 119, Why the mineral N before incubation was measured and not after one

week in the potential N mineralization method, ideally mineral N can be subtracted after 1 week of incubation. Since this time frame is used to enumerate the biota activity at optimal temperature and moisture content. Therefore, N mineralized during this time would be low and if this deleterious effects would not be adjusted then this effect may lead to underestimation of N mineral from the soil (Bloem et al. 1994).

In here, our purpose was to compare the difference between tillage systems rather than to obtain the absolute real value of soil N mineralization. Since the same test method was used for all tillage systems, errors or biases caused by the test method would be the same for samples collected from different tillage systems. But, we agree with the reviewer's suggestion that the activity of soil organisms may reduced after 4 weeks incubation. So, we used the inorganic nitrogen content measured in fresh soil every month instead of this amount of mineralized N obtained through lab incubation to indicate the status of soil N during soybean growth.

5.Line 148, please add the soil layer in cm where microarthropods were extracted? If the soil sample were collected from 15 cm soil depth, from the current unit it is not clear whether these organisms were extracted from 0-15 cm soil layer or 0-7.5 cm. How their contribution would be related to actual soil N mineralization from 0-5 and 5-15 cm? Although biota biomass from table 4 indicates the presence of these organisms in 0-5 and 5-15 cm, this should be explained in the methodology, in which depth actually the organisms were extracted.

The soil depths that soil organisms extracted from were added in the revised manuscript.

6.Earthworms were not present in this system or these organisms were not sampled from this experiment. Most of the published studies indicated their significant contribution after bacteria and fungi to N mineralization, it would be good to include their contribution in such systems.

The density of earthworms is less than 4 individual m-2 and their fresh weight is less

than 0.2 g individual-1 m-2 across all tillage systems. So, considering the low density and very small weight of earthworms in the studied region, we did not include them in this study.

7.Table S3, Why actual C:N ratio of the root of the soybean crop studied was not used. The currently used C:N ratio of soybean root is much less than the actual C:N ratio of the soybean, see for example (Kushwah et al. 2014; Redin et al. 2018). Such lower C:N ratio used in the calculation could lead to high N mineralization and hence overestimation of N mineralization in this category.

The C:N ratio of root in the literature (Kushwah et al. 2014; Redin et al. 2018) is based on the dry mass filled with the cellulose and lignin. But cellulose and lignin are not the main food for herbivoures. For example, plant-parasite nematodes primarily feed on the cytoplasm of root cells (Verschoor et al., 2002). So, using the actual C:N ratio of the soybean will underestimate the contribution of soil organisms to N mineralization. In our study, we used the C:N of the cytoplasm of root cells to indicate the C:N of root.

8.Table S4, values of biotic biomass were expressed in mg C m-2, but I could not find the reference of Berg et al. (1998) to confirm the average C content of 48% dry biomass used for microarthropods.

You can find the following sentences in the part of material of Berg et al. (1998): "The C content was set at 47.7% C for Acarida (Teuben 1991), —, and 47.5% C of the total dry weight for Collembola (Teuben 1991)."

9.For the nematodes biomass C, Ferris (2010) also adjusted this 0.1 C factor by using the formula, Pt= 0.1 Wt/mt, where Pt, Wt and mt are the C used in production, the body weight, and the cp class of taxon t, respectively. However, these factors may also influence the C biomass which may lead to over/underestimation of the biomass and therefore N mineralization by this group of soil biota.

Ferris (2010) used the formula: Pt= 0.1 Wt/mt to calculate the production C. Please
be note that the production C in not equal to the biomass C. The production C corresponds to the respiration C, which were both used to calculate the metabolic footprints of nematodes. So, the definitions of production C, respiration C and biomass C are very different.

10.The table S5, the contribution of N mineralization by a different group of soil organisms, these result are the main result according to the objective of the study, therefore can be moved to the main manuscript.

Thanks for your suggestion, we have reorganized the tables and figures in the revision as per your suggestion.

11.Line 163, please add the data about the number of taxa or abundance of soil organisms (nematodes and microarthropods in supplementary information or main text).

Thank you for your suggestion. The identified taxon was added as the supplementary information. The biomass of the identified taxa was moved to the main text..

12.Line 250, No difference in soybean yield among different treatments might be linked with the applied N fertilizer dose and/or atmospheric N deposition. Moreover, can you please explain why the difference in soil N mineralization among different treatments would not result in yield increment of soybean among different treatments? It seems yield was tended to be higher but did not differ significantly among treatments. Would the presentation of crop N uptake rather than crop yield explain the difference?

Just as the above what we said, there is no difference between the N input to soil among there tillage systems. So, it is unlikely that the high yield in conservation tillage, especially in NT, is related with the N fertilizer or atmospheric N deposition. The point is that, the total N mineralization of soil during the soybean growing season is distributed unevenly throughout the plow layer under conservation tillage systems. The amount of N mineralization at 0-5 cm was higher in RT and NT than in CT; but, the opposite trend was observed in 5-15 cm (Fig. 1). This poses a question, how did the deficit of

mineral N in 5-15 cm support the high yield in RT and NT soils? We thought that the contribution of soil organisms to N mineralization may offset this disadvantage. And this is main reason why we calculated the N mineralization of the soil organisms in this study.

13.Fig. S1a, the P-values presented in the figure indicate that tillage, depth, and their interaction were significant, please use multiple comparisons to differentiate the effects, if done already please add letters on the bar to differentiate the effect of the treatments within or between the two depths. These are the main result, therefore, I suggest presenting them in the main manuscript rather than supplementary information.

Thank you for your suggestion, this was done in the revisoin

14.Line 251, presenting the biomass or abundance data in the main manuscript would add more value, therefore I would suggest adding this data in the manuscript.

Thank you for your suggestion, this was done in the revision.

15.Line 263, indicate that bacterivorous nematodes and omnivorous-predaceous ne-matodes contributed highest to N mineralization that was not the case in Table S5. Can you please discuss this difference in detail in the discussion section? Or I could not understand from the current formulation what do you mean?

These sentences were rewritten. What we want to describe is that the relative changes were most pronounced in bacterivorous nematodes and omnivorous-predaceous ne-matodes, their contribution to N mineralization was greatly improved under RT and NT soils.

16.Fig. 1 The response ratio of soil N mineralized during the growing season and crop yield was calculated, is that a fair comparison. Is it not better to use the response ratio of soil N mineralization and crop N uptake? To check for the accuracy of modeling: did the temporal variation in calculated N-mineralization rates correspond with the temporal variation in measured N-mineralization rates (potential N mineralization)? I could not

see this in the manuscript. The main aim of the manuscript is to examine the influence of soil biota on coupling N mineralization with soybean yield therefore the current fig. 1 did not meet the objective. Hence, I would suggest to also include the response ratio of soil N calculated based on the modeling and soybean yield.

The description of response ratio was deleted in the revised manuscript.

17.Fig. 2, what is the difference between mineralization N delivered by soil biota and of the contribution of soil biota to soil mineralization N? Please clarify it.

Their units are different. For the mineralization N delivered by soil biota, the unit is expressed as kg N ha-1; for the contribution of mineralization N of soil biota to soil mineralization N, the unit is dimensionless based on standardization. The contribution of mineralization N of soil biota to soil mineralization N was deleted in the manuscript to make the text more clear to readers.

18.Discussion Line 285, In the case of Holtkamp et al. (2011) bacteria and fungi contributed about 77% of the total N mineralized which is in line with Rashid et al. (2014), who estimated that the aforementioned biota contributed to the 60% of the soil N mineralized. So, bacteria and fungi but not the higher trophic groups were responsible for most of the soil N mineralization in their systems. Even in your system Table S5, the contribution of fungi is the highest followed by bacteria and there is an insignificant contribution to N mineralization is coming from nematodes and microarthropods. What do you mean by the higher trophic group here?

We want to express that "For the mineral N delivered by soil organisms, the differences between CT and RT and NT were more pronounced for higher trophic groups than for lower trophic ones". It is not our purpose to compare the amount of mineralization N of soil organisms or the contribution of mineralization N of soil organisms to soil N mineralization among different trophic groups. Our focus on the comparison between different tillage systems, because these differences among tillage systems may be the primary reason of soil N mineralization and plant yield differences.

19.Lines 328-330, why fungal pathways were dominated in the soil layer 0-5 and bacterial pathways in the layer 5-15 cm in RT and NT tillage? Can you please mechanistically explain how these pathways contributed to soybean yield? In lines 335-341, I expected the discussion on why the fungal pathways were dominated contributors of soybean yield in 0-5 cm and bacterial pathways in 5-15 cm soil layer? Can you please discuss further how and why these pathways were dominated in these layer under RT and NT tillage operations.

We reconstructed the soil food web and calculated the mineral N delivered by soil organisms, and then found that RT and NT mainly drive the N mineralization through fungal and bacterial channels at the whole plow layer (0-15 cm). But, when use stepwise regression analysis to relate the N mineralization of different channels with soybean yield, the results have shown that at 0-5 cm, fungal channel was significantly related with soybean yield, while at 5-15 cm, bacterial channel was strongly related with soybean yield. These results suggest that the different pathways of soil organisms relating N mineralization and plant yield.

20.The manuscript uses modeling to estimate various fluxes of N in the soybean. In the model, a lot of parameters were taken from literature rather than from measurements in the actual sites. What the authors fail to discuss (and to mention), is that there is a degree of uncertainty associated with any model. Each estimate based on modelling equations comes with the error range. Depending on the model and the parameter in question, this error range can be small or large. Therefore, a sensitivity analysis should be carried out. Moreover, it needs to be mentioned, if any conclusions are to be drawn based on model-derived numbers. A model estimate for any parameter should never be presented as a single number without an error range. I encourage the authors to reflect this in the Discussion and Conclusion. Please provide the error range for the values you estimate based on models, and please adjust your Discussion of differences in soil N fluxes, and your Conclusions, to reflect the uncertainties associated with modeling.

Thanks for your suggestion. The soil food web was rebuilt in the revised manuscript.

Furthermore, we re-calculated the N mineralization of soil organisms according to Ruiter et al. (1993). Sensitivity analysis was conducted to test the influence of the uncertainty on the result of N mineralization. All ambiguous results were deleted, and the discussion was rewritten to obtain a concise and logical conclusion.

References 1.Berg, M.P, Kniese, J.P., Bedaux, J.J.M., Verhoef, H.A.: Dynamics and stratification of functional groups of micro- and mesoarthropods in the organic layer of a Scots pine forest, Biology and Fertility of Soils, 26, 268-284, 1998. 2.de Ruiter, P.C., Van Veen, J.A., Moore, J.C. Brussaard, M.L. and Hunt, H.W.: Calculation of nitrogen mineralization in soil food webs, Plant & Soil, 157, 263-273, 1993. 3.Ferris, H.: Form and function: Metabolic footprints of nematodes in the soil food web, European Journal of Soil Biology, 46, 97–104, 2010. 4.Gómez-Rey, M.X., Couto-Vázquez, A., González-Prieto, S.J.: Nitrogen transformation rates and nutrient availability under conventional plough and conservation tillage, Soil & Tillage Research, 124, 144-152, 2012. 5.Verschoor, B.C.: Carbon and nitrogen budgets of plant-feeding nematodes in grasslands of different productivity, Apply Soil Ecology, 20, 15-25, 2002. 6.Wienhold, B.J.: Comparison of laboratory methods and an is situ method for estimating nitrogen mineralization in an irrigated silt-loam soil, Comminications in Soil Science and Plant Analysis, 38, 1721-1732, 2007.

[Figure]

Fig. 1 Cumulative soil mineral N concentration during soybean growing season under different tillage practices (mean ± standard error). Tillage practices capped by the same lowercase letter are not significantly different ($P > 0.05$). CT, conventional tillage; RT, ridge tillage; NT, no tillage.

**Fig. 1.**

---

## Author Comment (AC3) · 26 May 2020

1.Specific comments One key concern is that N mineralization is measured under laboratory conditions and then corrected to field conditions, via a solely temperaturedependent Q10 equation (L112-114). It is well known that the simple Q10 relationship does not hold under realistic soil conditions, since temperature is not the only limiting factor. Soil moisture, substrate availability, etc also strongly co-determine the biogeochemical process rates in situ (see e.g. Davidson & Jansses 2006 Nature 440: 165-173 for SOM decomp). Therefore, I do not believe that the authors can capture realistic N mineralization rates in their field. I think this paper needs a thorough validation of this

relationship.

The lab incubation method and the in situ method are the most common method used in the researches to investigate the soil N mineralization rate. But, both methods have their own limitations (Hanselaman et al., 2004; Wienhold, 2007). So, obtaining the absolute real value is impossible. We used the inorganic nitrogen content measured in fresh soil every month instead of this amount of mineralized N obtained through lab incubation to indicate the status of soil N under different tillage systems during soybean growth. What we want to emphasize is that obtaining absolute value was not our goal, making a comparison between different farming systems, specifically, tillage systems was our core objective. Since the same test method was used for all tillage systems, errors or biases caused by the test method would be the same for samples collected from different tillage systems.

2.Similarly, I am highly critical of the way the authors attribute N mineralization contributions from different soil biota groups. They use a series of equations from other authors to transform soil biota abundances into process rates (e.g. L170-L176, L177-188, L198-202). Mostly these steps seem to be based on Rashid et al 2014. These steps form the heart of their study. For instance, the conclusion that conservation tillage promotes N min (L21-23), hinges on these equations that all assume that more soil biota lead to more N min. The same goes for the relative contributions of soil biotic groups to total N mineralization (L25-27). The parameter estimates (e.g. Q10 of 3, L116) used come from different systems in other countries, while it is know that N cycling processes are highly heterogeneous in space and time. I am therefore sceptical that the same relations and the same parameter estimates will hold in the system studied by the authors. In fact even in the source paper, Rashid et al 2014, the ecologicalproduction model is an improvement over the standard government rules, but still there is considerable error in the estimates (87-120% of observed N min rates) on the fields they studied. So I think the authors have to spend much more effort on convincing me and other readers that using these equations leads to valid inferences

about this particular system. To be honest, as an empiricist, I think that to only realistic way to get to these questions is to use isotopic tracers in the field plots. However, what would help is if 1) we had realistic data on N min rates in the actual plots, and 2) the summed N contributions over the soil biota would have a strong predictive relationship with these independent field data. As it stands such a field validation is totally missing, which makes the study unconvincing.

Researchers have using the theoretical method to quantify the elemental energy flux of soil food webs for more than thirty years. The parameters, such as assimilation efficiency, the ratio of C:N of predator or prey, and feeding preference and so on, used in this method were almost constant over the past thirty years. The classic literature is de Ruiter et al. (1993), Didden et al. (1994) and Hunt et al. (1987), and the recent literature of Andrés et al. (2016, Soil Biology and Biochemistry), de Vries et al. (2013, PNANS) and Schwarz et al. (2017, Nature Climate Change) also used this method to explore the C or N flow through soil food web in the grassland ecosystem of America, agroecosystem of Europe and the forest ecosystem of America. The method is well established and accepted by researchers. So far, as far as we know, there is no research using this theoretical method to quantify the energy flux of the soil food web in Asia or China. In the revised manuscript, we re-calculated the N mineralization of soil organisms according to Ruiter et al. (1993).

3.Data were missing in some months for nematode data and linear interpolation was used to fill these data gaps (L129). I find this a risky approach, especially since nematode population dynamics within season are non-linear, see e.g. the data in Rashid et al, but also other sources. I think the authors also need to show that their conclusions hold if the only work with the months where they have data on all soil groups.

The nematode populations for non-sampled months were estimated by linear interpolation between adjacent sampling dates. This method is usually used in the literature (Didden et al., 1994; Berg et al., 2001; Zhang et al., 2019), which assumed that there is a linear course in biomass or abundance of soil organisms between sampling dates.

This method can not track the trend of nematodes population changes, but can yield a more accurate mean value during the studied period.

4.The authors use the ratios of (calculated) mineral N delivery in the conservation tillage (ridge, and no tillage) to conventional tillage in their main figures. However, ratios are biased (e.g. Jasienski & Bazzaz 1999 Oikos 84: 321-326); a log(Treat/Control) has better statistical properties (Brinkman et al 2010 J Ecol 98: 1063–1073). Even better however would be if the main analyses and figures are directly based on the data from the three treatments directly, this approach would even give you a bit more statistical power. In that sense I find the supplementary figures to be much clearer.

Thank you for your suggestion. The tables and figures were reorganized in the revised manuscript.

5.In general, I find that the writing is a bit to colloquial in tone and imprecise in many places. See some examples below. Also I find that the presentation of the energy channels to be a bit overstated, there have been many findings of cross-feeding across these 'channels', and really I think we need to adopt a network view of the soil community and its links to biogeochemical processes.

The soil food web was rebuilt in the revised manuscript. Sensitivity analysis was conducted to test the influence of the uncertainty on the result of N mineralization. All ambiguous results were deleted, and the discussion was rewritten to obtain a concise and logical conclusion.

6.Minor comments - L44: what do you mean with 'special species'? - L51: what are weak root infections - L55: what do you mean by capacity? Use of substrates? Process rates? - L60: I would not use the word conquer here, maybe mediate? - L61: adverse effects on what? - L66: rich in what sense - L68: what is stratified and in what way? - L80: based on M&M I believe its 14 years, not 15. - L83: what do you mean coupling? How will you quantify that coupling? - L85: it is a bit unclear what you mean by multiple spatial interactions in this hypothesis. How will you test this? - L94: how big were the

plots? - L100: what was done with the maize residue?

These inappropriate points were rewritten and the missed information was added to the revised manuscript.

References 1.Andrés, P., Moore, J.C., Simpson, R.T., Selby, G., Cotrufo, F., Denef, K., Haddix, M.L., Shaw, E.A., de Tomasel, C.M., Molowny-Horas, R., and Wall, D.H.: Soil food web stability in response to grazing in a semi-arid prairie: The important of soil textural heterogeneity, Soil Biology & Biochemistry, 97, 131-143, 2016. 2.Berg, M., de Ruiter, P., Didden, W. Janssen, M., Schouten, T. and Verhoef, H.: Community food web, decomposition and nitrogen mineralisation in a stratified Scots pine forest soil, Oikos, 94, 130-142, 2001. 3.de Ruiter, P.C., Van Veen, J.A., Moore, J.C. Brussaard, M.L. and Hunt, H.W.: Calculation of nitrogen mineralization in soil food webs, Plant & Soil, 157, 263-273, 1993. 4.de Vries, F.T., Thébault, E., Liiri, M. Birkhofer, K., Tsiafouli, M.A., Bjørnlund, L., Jørgensen, H.B., Brady, M.V., Christensen, S., de Ruiter, P. C., d'Hertefeldt, T., Frouz, J., Hedlund, K., Hemerik, L., Gera Hol, W.H., Hotes, S., Mortimer, S.R., Setälä, H., Sgardelis, S.P., Uteseny, K., van der Putten, W.H., Wolters, V. and Bardgett, R.D.: Soil food web properties explain ecosystem services across European land use systems, Proceedings of the National Academy of Sciences, 110, 14296-14301, 2013. 5.Didden, W.A.M., Marinissen, J.C.Y., Vreeken-Buijs, M.J. Burgers, S.L.G.E., de Fluiter, R., Geurs, M. and Brussaard, L.: Soil meso- and macrofauna in two agricultural systems: factors affecting population dynamics and evaluation of their role in carbon and nitrogen dynamics, Agriculture, Ecosystems & Environment, 51, 171-186, 1994. 6.Halpern, M.T., Whalen, J.K., Madramootoo, C.A.: Long-term tillage and residue management influences soil carbon and nitrogen dynamics, Soil Science Society of America Journal, 74(4), 1211-1217, 2010. 7.Hanselman, T.A., Graetz, D.A., Obreza, T.A.:A comparison of in situ methods for measuring net nitrogen mineralization rates of organic soil amendments, Journal of Environmental Quality, 33, 1098-1105, 2004. 8.Hunt, H.W., Coleman, D.C., Ingham, E.R. Ingham, R.E., Elliott, E.T., Moore, J.C., Rose, S.L., Reid, C.P.P. and Morley, C.R.: The detrital food web in a

shortgrass prairie, Biology & Fertility of Soils, 3, 57-68, 1987. 9.Schwarz, B., Barnes, A.D., Thakur, M.P., Brose, U., Ciobanu, M., Reich, P.B., Rich, R.L., Rosenbaum, B., Stefanski, A and Eisenhauer, N.: Warming alters energetic structure and function but not resilience of soil food webs, Nature Climate Change, 7, 895–900, 2017. 10.Zhang, S.X., McLaughlin, N.B., Cui, S.Y., Yang, X.M., Liu, P., Wu, D.H. and Liang, A.Z.: Effects of long-term tillage on carbon partitioning of nematode metabolism in a Black soil of Northeast China, Applied Soil Ecology, 138, 207-212, 2019.

---

## Author Comment (AC4) · 26 May 2020

1.However, both reviewers raise concerns with how models from the literature were applied to this specific study. Applying models can offer insights and predictions, but it is important to understand and report the uncertainties that arise from inputting field and laboratory data from one study into a model developed in another. A way to address this would be to conduct a sensitivity test, as suggested by reviewer 1. Additionally, caveats need to be incorporated throughout the results and discussion, especially to the conclusions, which both reviewers felt were overstating the underlying data.

Thank you for your suggestion. We are very appreciative of the reviewers' suggestions

to obtain the reliable results and discussion. We rebuilt the soil food web based on the trophic relationship among microbes, nematodes, collembolans and mites. And then these trophic groups were classified into six feeding guilds: bacteria, fungi, bacter-vorous feeders, fungivorous feeders, herbivorous feeders and predators. The N mineralization of soil food web were also re-calculated according to Ruiter et al. (1993). And according to the suggestion of reviewer1's suggestion, the sensitivity analysis was conducted to test the impact of uncertainty of the model on the result of N mineralization. In addition, as we reanalyzed, the results and the discussion were also rewritten. According to the reviewer's suggestion, all inappropriate views were deleted.

2.Both reviewers mention ways the text could be improved for clarity. In some cases, there is confusion around methods, which may require some extensive rewriting. It is important to consider where grammatical changes can improve the text and where additional information is truly required. A third reviewer found the writing too confusing to do a full reviewer; however, the thorough review of the other two reviewers provides sufficient feedback to proceed with revision of the manuscript.

We reorganized the structure of this manuscript to make it clear and concise to readers. We also invited the native English speaking researcher to polish this manuscript. We believe that the revised manuscript will be satisfactory.

References 1.de Vries, F.T., Thébault, E., Liiri, M. Birkhofer, K., Tsiafouli, M.A., Bjørn-lund, L., Jørgensen, H.B., Brady, M.V., Christensen, S., de Ruiter, P. C., d'Hertefeldt, T., Frouz, J., Hedlund, K., Hemerik, L., Gera Hol, W.H., Hotes, S., Mortimer, S.R., Setälä, H., Sgardelis, S.P., Uteseny, K., van der Putten, W.H., Wolters, V. and Bardgett, R.D.: Soil food web properties explain ecosystem services across European land use systems, Proceedings of the National Academy of Sciences, 110, 14296-14301, 2013.
* * *

---

## Author Response (AR1)

Dr. Elizabeth Bach
Topical Editor
SOIL

Dear Editor,

Re: Manuscript soil-2020-2 "Multi-cooperation of soil biota in the plough layer is the key for conservation tillage to improve N availability and crop yield" Shixiu Zhang et al.

Thank you very much for your careful review and constructive suggestions with regard to our manuscript. We sincerely appreciate the reviewer's and Editor's thorough reviews and helpful suggestions. I am sending here one copy of our revised manuscript, with the revised portion marked in blue, a file with highlights, and a revised appendix file.

The responses to the reviewer's and Editor's comments are listed below.

We believe that we have addressed all of the reviewers comments and that manuscript has been improved satisfactorily. We hope it will meet your approval.

Yours sincerely,

Dr. Shixiu Zhang

**Topical Editor**

Comments from the reviewers are in normal font and our responses are marked in blue. The line references in our responses refer to the line numbers in the revised manuscript.

1. However, both reviewers raise concerns with how models from the literature were applied to this specific study. Applying models can offer insights and predictions, but it is important to understand and report the uncertainties that arise from inputting field and laboratory data from one study into a model developed in another. A way to address this would be to conduct a sensitivity test, as suggested by reviewer 1. Additionally, caveats need to be incorporated throughout the results and discussion, especially to the conclusions, which both reviewers felt were overstating the underlying data.

Thank you for your suggestion. We are very appreciative of the reviewers' suggestions to obtain the reliable results and discussion. We extensively revised the manuscript to address the criticisms and shortcomings raised by the reviewers. We reconstructed the soil food web based on the trophic relationship among microbes, nematodes, collembolans and mites. And then these trophic groups were classified into six feeding guilds: bacteria, fungi, bacterivorous feeders, fungivorous feeders, herbivorous feeders and predators. The N mineralization of soil food web was also re-calculated according to de Ruiter et al. (1993). And according to the suggestion of reviewer1's suggestion, a sensitivity analysis was conducted to test the impact of uncertainty of assignment of feeding preferences of omnivorous collembolans in the model on the N mineralization results.

In addition, as we reanalyzed, the results and the discussion were also rewritten. According to the reviewer's suggestion, all inappropriate views were deleted.

2. Both reviewers mention ways the text could be improved for clarity. In some cases, there is confusion around methods, which may require some extensive rewriting. It is important to consider where grammatical changes can improve the text and where additional information is truly required. A third reviewer found the writing too confusing to do a full reviewer; however, the thorough review of the other two reviewers provides sufficient feedback to proceed with revision of the manuscript.

To make the text clear and concise to readers, we reorganized the structure of this manuscript and rewrote the abstract, introduction, results and discussion, and explained the material and methods in detail. We also invited the native English speaking researcher to polish this manuscript. We believe that the revised manuscript will be satisfactory.

The content marked in blue in the revision is for the convenience of tracking the modification according to the reviewers' suggestion. But, other contents, such as abstract, introduction, results, discussion and conclusion, etc., were also revised intensively .

**Reviewer 1**

1. Material and Methods What was the motive behind choosing 0-5 and 5-15 cm soil layer for soil biota sampling and N mineralization when the plow layer for conventional tillage was 20 cm? For the latter case, tillage operation mixed the soil layer of 0-20 cm. Why bulk density was recorded at 5 cm and 10 cm and not 0-5 and 5-15 cm soil depth? The difference in bulk density might affect the soil N mineralization.

Soil stratification is a typical characteristic of conservation tillage, because there is a contrasting difference between top soil (usually means 0-5 cm) and the sub soil. Using either 5-15 cm or 5-20 cm to investigate the conservation tillage effect on the sub-soil depth is very common in the literature (for example, 5-15 cm in the study of Gómez-Rey et al., 2012; 5-20 cm in Haplern et al., 2010). Our previous study found that there was no significant difference between these two soil depths (5-15 cm and 5-20 cm) in soil C, N, bulk density, soil water content, and the other soil physicochemical parameters, but there was a slight difference in the abundance of soil collembolans and mites. Their abundance at the 20 cm depth was very low. So, on this basis, we think it is more reasonable to use 5-15 cm to investigate the role of soil organisms.

Lines 142-143: We rewrote the description in the paper about how the soil bulk density was determined. The soil cores for the bulk density were 0-5 cm and 5-15 cm, and therefore, the bulk density data obtained represent the mean of these two depth bands, not 5 cm and 10 cm.

2. Line 95, Zhang et al. (2019) used 40 kg N ha-1 in the soybean field. Moreover, there might be atmospheric N deposition. Therefore, all or part of the N from the applied fertilizer and/or atmospheric N deposition can be taken up by soybean and help to increase the yield of soybean, how this effect of N fertilization on crop N uptake/yield was separated from N contribution by a different trophic group of soil organisms? Please explain why N fertilizer (40 kg N ha-1) plot wAS

considered as a suitable reference to estimate background crop yield/N response.

We focused on investigating the difference of N mineralization by soil organisms among different tillage systems, not on the crop yield response to soil N input rate. Furthermore, the amount of soil input N as fertilizer was the same in all tillage systems; the amount of N fertilizer applied is 2/3 of the typical amount for soybean grown by local farmers. For the deposition of atmospheric N, its contribution can be neglected even if it is not uniformly distributed in the atmosphere, because it is very small relative to the amount of nitrogen fertilizer applied; further, all plots in the experimental site would receive the same deposition from the atmosphere. Therefore, in this context, there would be no significant difference in the utilization of applied N in soybean of the same variety.

We deemed it necessary to discuss whether soil organisms play a key role in N mineralization as nitrogen fertilizer application is reduced. As discussed in line 362-380 of the revision, N supply from soil mineral N is insufficient to achieve maximum soybean yield and must be supplemented by the N release from soil organisms.

3. Line 107, please add the soil depth at which temperature was recorded.

These sentences described the determination of soil temperature were deleted from the revised manuscript because soil temperature was not used in the calculation of N mineralization of soil organisms according to the equation provided by de Ruiter et al. (1993).

4. Line 119, Why the mineral N before incubation was measured and not after one week in the potential N mineralization method, ideally mineral N can be subtracted after 1 week of incubation. Since this time frame is used to enumerate the biota activity at optimal temperature and moisture content. Therefore, N mineralized during this time would be low and if this deleterious effects would not be adjusted then this effect may lead to underestimation of N mineral from the soil (Bloem et al. 1994).

Our purpose in the experiment was to compare the difference between tillage systems rather than to obtain the absolute real value of soil N mineralization. Since the same test method was used for all tillage systems, errors or biases caused by the test method would be the same for samples collected from different tillage systems.

But, we agree with the reviewer's suggestion that the activity of soil organisms may be lower averaged over 4 weeks incubation than that if they were allowed to stabilize for a week prior to initial measurement. So, in the revision, we used the inorganic nitrogen content measured in fresh soil samples obtained from the field every month instead of the amount of mineralized N obtained through lab incubation (the method used in the original submission) to indicate the status of soil N during soybean growth; this avoids the problem raised by the reviewer. Field sampling was described in lines 138-140 and the method of determining soil mineral N was presented in the lines 146-148 of the revision.

5.  Line 148, please add the soil layer in cm where microarthropods were extracted? If the soil sample were collected from 15 cm soil depth, from the current unit it is not clear whether these organisms were extracted from 0-15 cm soil layer or 0-7.5 cm. How their contribution would be related to actual soil N mineralization from 0-5 and 5-15 cm? Although biota biomass from table 4 indicates the presence of these organisms in 0-5 and 5-15 cm, this should be explained in the methodology, in which depth actually the organisms were extracted.

The soil depths that soil organisms extracted from were added in the revised manuscript in line 155-157. It was stated in lines 138-140 that the field soil samples were cores from the 0-15 depths, and the cores were separated into 0-5 and 5-15 cm depths, and subsamples from each depth were combined to form a single composite sample at each depth for each plot. This method of taking one deep core and separating the core into sections corresponding to two or more depths is widely used in field research to examine the effect of depth on a measured parameter; it avoids potential contamination by sloughing of the core sidewalls which can occur if a 0-5 cm core is first removed, and the coring probe reinserted into the same hole to get a separate 5-15 cm core. From this information, it is evident that the extracted organisms represent the mean of the entire range of the depths of the separated cores, 0-5 cm, and 5-15 cm.

6.  Earthworms were not present in this system or these organisms were not sampled from this experiment. Most of the published studies indicated their significant contribution after bacteria and fungi to N mineralization, it would be good to include their contribution in such systems.

The density of earthworms is less than 4 individual $m^{-2}$ and their fresh weight is less than 0.2 g $individual^{-1}$ across all tillage systems. So, considering the low density and very small weight of earthworms in the studied region, we did not include them in this study.

7.  Table S3, Why actual C:N ratio of the root of the soybean crop studied was not used. The currently used C:N ratio of soybean root is much less than the actual C:N ratio of the soybean, see for example (Kushwah et al. 2014; Redin et al. 2018). Such lower C:N ratio used in the calculation could lead to high N mineralization and hence overestimation of N mineralization in this category.

The C:N ratio of root in the literature (Kushwah et al. 2014; Redin et al. 2018) is based on the dry mass which contains a high portion of cellulose and lignin. But cellulose and lignin are not the main food for herbivores. For example, plant-parasite nematodes primarily feed on the cytoplasm of root cells (Verschoor et al., 2002). So, using the actual C:N ratio of the soybean root will underestimate the contribution of soil organisms to N mineralization. In our study, we used the C:N of the cytoplasm of root cells to indicate the C:N of root. This was clarified in Table S4 of the revision; the source of the C:N values is given in Table S4.

8. Table S4, values of biotic biomass were expressed in mg C m-2, but I could not find the reference of Berg et al. (1998) to confirm the average C content of 48% dry biomass used for microarthropods.

Sorry, on checking, we realized that we had cited Berg et al. (1998) in the footnote for Table S4 (original submission), but we had neglected to include Berg et al. (1998) in the references. Thank you for pointing this out. The same information is in Berg et al. (2001) so we only cited this paper and did not cite Berg et al. (1998) in the revision.

You can find the following sentences in the part of material of Berg et al. (1998): "The C content was set at 47.7% C for Acarida (Teuben 1991), ---, and 47.5% C of the total dry weight for Collembola (Teuben 1991)."

9. For the nematodes biomass C, Ferris (2010) also adjusted this 0.1 C factor by using the formula, Pt= 0.1 Wt/mt, where Pt, Wt and mt are the C used in production, the body weight, and the cp class of taxon t, respectively. However, these factors may also influence the C biomass which may lead to over/underestimation of the biomass and therefore N mineralization by this group of soil biota.

Ferris (2010) used the formula: Pt= 0.1 Wt/mt to calculate the production C. Please note that the production C is not equal to the biomass C. The production C is defined as the C used for anabolism (Zhang et al., 2019); the biomass C is the C contained within the living component of soil organisms.

10. The table S5, the contribution of N mineralization by a different group of soil organisms, these result are the main result according to the objective of the study, therefore can be moved to the main manuscript.

Thanks for your suggestion, we have reorganized the tables and figures in the revision as per your suggestion. The material in Table S5 in the original submission has been reformatted and moved to Table 3 in the revision.

11. Line 163, please add the data about the number of taxa or abundance of soil organisms (nematodes and microarthropods in supplementary information or main text).

Thank you for your suggestion. The identified soil organism taxa were added as the supplementary information in Tables S1 and S2. The biomass of the identified taxa was moved to the main text..

12. Line 250, No difference in soybean yield among different treatments might be linked with the applied N fertilizer dose and/or atmospheric N deposition. Moreover, can you please explain why the difference in soil N mineralization among different treatments would not result in yield increment of soybean among different treatments? It seems yield was tended to be higher but did not differ significantly among treatments. Would the presentation of crop N uptake rather than crop yield explain the difference?

We agree that the N applied as fertilizer would have some effect on diluting the yield response to tillage, i.e. typical flattening of the yield N response curve at higher total N rates. However, the difference among tillage systems in total N mineralized by soil organisms (Table 3 in the revision) was over twice the applied amount, and we attributed the difference in yield to the difference in mineralized N. The yield followed the order NT > RT > CT, and although not significant, the trend was consistent with the differences in mineralized N among the tillage systems, NT > RT > CT (Table 3 in the revision). We attribute the lack of significance to normal within experiment variability.

As discussed in our response to comment 2 above there was no difference between the N input to soil (fertilizer + atmospheric deposition) among the three tillage systems. So, it is unlikely that the higher yield in conservation tillage, especially in NT, was related to the N fertilizer or atmospheric N deposition. The point is that, the amount of soil mineral N during the soybean growing season is distributed unevenly throughout the plow layer under conservation tillage systems. The amount of soil mineral N at 0-5 cm was higher in RT and NT than in CT; but, the opposite trend was observed in 5-15 cm (Table 1 in the revision). This poses a question, how did the deficit of mineral N in 5-15 cm support the higher yield in RT and NT soils? We surmised that the contribution of soil organisms to N mineralization may offset this disadvantage. And this is main reason why we calculated the N mineralization of the soil organisms in this study.

Line 362-380: we rewrote the discussion to clarify why the mineralizable N mediated by soil organisms rather than the inherent soil mineral N plays a key role in meeting the requirements of plant growth in RT and NT soils. The soil organisms are producing mineralized N as it is being used by the plant, and thus, the yield in NT and RT was higher than CT even though the mineral N remained low in RT and NT soils.

13. Fig. S1a, the P-values presented in the figure indicate that tillage, depth, and their interaction were significant, please use multiple comparisons to differentiate the effects, if done already please add letters on the bar to differentiate the effect of the treatments within or between the two depths. These are the main result, therefore, I suggest presenting them in the main manuscript rather than supplementary information.

Thank you for your suggestion, this was done in the revision. The material in Fig. S1 in the original submission is now included in Table 1 in the revision, and the letters indicating pairwise differences are included.

14. Line 251, presenting the biomass or abundance data in the main manuscript would add more value, therefore I would suggest adding this data in the manuscript.

Thank you for your suggestion, this was done in the revision.

15. Line 263, indicate that bacterivorous nematodes and omnivorous-predaceous nematodes contributed highest to N mineralization that was not the case in Table S5. Can you please discuss this difference in detail in the discussion section? Or I could not understand from the current formulation what do you mean?

These sentences were rewritten. We reassigned the identified soil organisms (bacteria, fungi, nematodes, mites and collembolans) into six functional feeding guilds: bacteria, fungi, herbivorous feeders, bacterivorous feeders, fungivorous feeders, and predaceous feeders. And then their contributions to N mineralization were recalculated.

16. Fig. 1 The response ratio of soil N mineralized during the growing season and crop yield was calculated, is that a fair comparison. Is it not better to use the response ratio of soil N mineralization and crop N uptake? To check for the accuracy of modeling: did the temporal variation in calculated N-mineralization rates correspond with the temporal variation in measured N-mineralization rates (potential N mineralization)? I could not see this in the manuscript. The main aim of the manuscript is to examine the influence of soil biota on coupling N mineralization with soybean yield therefore the current fig. 1 did not meet the objective. Hence, I would suggest to also include the response ratio of soil N calculated based on the modeling and soybean yield.

The description of response ratio and the material in Fig. 1 was deleted in the revised manuscript. The focus of the revision was changed to differences among tillage systems in N mineralization by the different organism feeding guilds and the subsequent effect on soybean yield.

17. Fig. 2, what is the difference between mineralization N delivered by soil biota and of the contribution of soil biota to soil mineralization N? Please clarify it.

Their units are different. For the mineralization N delivered by soil biota, the unit is expressed as kg N ha$^{-1}$; for the contribution of mineralization N of soil biota to soil mineralization N, the unit is dimensionless based on standardization.

The contribution of mineralization N of soil biota to soil mineralization N was deleted in the manuscript to make the text more clear to readers.

18. Discussion Line 285, In the case of Holtkamp et al. (2011) bacteria and fungi contributed about 77% of the total N mineralized which is in line with Rashid et al. (2014), who estimated that the aforementioned biota contributed to the 60% of the soil N mineralized. So, bacteria and fungi but not the higher trophic groups were responsible for most of the soil N mineralization in their systems. Even in your system Table S5, the contribution of fungi is the highest followed by bacteria and there is an insignificant contribution to N mineralization is coming from nematodes and microarthropods. What do you mean by the higher trophic group here?

It was not our purpose to compare the amount of mineralization N of soil organisms or the contribution of mineralization N of soil organisms to soil N mineralization among different trophic groups. Our focus was on the comparison among different tillage systems, because these differences among tillage systems may be the primary reason for soil N mineralization and plant yield differences. So, we deleted these unclear sentences in the revision.

19. Lines 328-330, why fungal pathways were dominated in the soil layer 0-5 and bacterial pathways in the layer 5-15 cm in RT and NT tillage? Can you please mechanistically explain how these pathways contributed to soybean yield? In lines 335-341, I expected the discussion on why the fungal pathways were dominated contributors of soybean yield in 0-5 cm and bacterial pathways in 5-15 cm soil layer? Can you please discuss further how and why these pathways were dominated in these layer under RT and NT tillage operations.

We reconstructed the soil food web and calculated the mineral N delivered by soil organisms, and then found that RT and NT mainly drive the N mineralization through fungal and bacterial channels at the whole plow layer (0-15 cm). But, when we used stepwise regression analysis to relate the N mineralization of different channels with soybean yield, the results showed that at 0-5 cm, fungal channel was significantly related with soybean yield, while at 5-15 cm, bacterial channel was strongly related with soybean yield. These results suggest that different soil organisms dominate at different depths in driving N mineralization and plant growth. This was clarified in the revision. In lines 396-407 in the revision, we discussed about dominance of fungi in the 0-5 cm depth because fungi can transfer nutrients from surface residue via hyphae.

20. The manuscript uses modeling to estimate various fluxes of N in the soybean. In the model, a lot of parameters were taken from literature rather than from measurements in the actual sites. What the authors fail to discuss (and to mention), is that there is a degree of uncertainty associated with any model. Each estimate based on modelling equations comes with the error range. Depending on the model and the parameter in question, this error range can be small or large. Therefore, a sensitivity analysis should be carried out. Moreover, it needs to be mentioned, if any conclusions are to be drawn based on model-derived numbers. A model estimate for any parameter should never be presented as a single number without an error range. I encourage the authors to reflect this in the Discussion and Conclusion. Please provide the error range for the values you estimate based on models, and please adjust your Discussion of differences in soil N fluxes, and your Conclusions, to reflect the uncertainties associated with modeling.

Thanks for your suggestion. The soil food web was rebuilt in the revised manuscript. Furthermore, we re-calculated the N mineralization of soil organisms according to Ruiter et al. (1993). Sensitivity analysis was conducted to test the influence of the uncertainty in the feeding preference of omnivorous collembolans on the result of N

mineralization and the modelling performance was also discussed in the line 318-342. All ambiguous results were deleted, and the discussion was rewritten to obtain a concise and logical conclusion.

**Reviewer 2**

1.  Title: needs re-working. "Multi-cooperation" isn't correct. Perhaps simply "interaction"?

The title of this manuscript was re-worked.

2.  Affiliations: I think there should be a better translation for "Key Laboratory of Mollisols Agroecology". Even simply "Laboratory of Mollisol Agroecology"

The translation of our organization is the official translation and cannot be modified. This name was adopted several decades ago.

3.  Ln 13: Please check English grammar. For example, "Conservation tillage systems may promote more complex and heterogeneous distributions of soil organisms relative to conventional tillage that may result in higher crop yield. However, the role of soil biota in N mineralization promoting plant growth remains limited."

Thank you for suggestion. We have invited a native English researcher to help revise the paper.

4.  Some introduction or definition of "trophic groups" and "energy pathways" is needed.

We reconstructed the soil food webs, and assigned the identified soil organisms (bacteria, fungi, nematodes, mites and collembolans) into six functional feeding guilds: bacteria, fungi, herbivorous feeders, bacterivorous feeders, fungivorous feeders, and predaceous feeders. Therefore, in the line 205-206, the definition of 'trophic feeding guild' was given in the revised manuscript.

5.  Ln 27-31: Is the second to last sentence of the Abstract the main finding of the study? The last statement, on lines 30 and 31, is quite a broad generalization and is not overly useful. The second to last sentence here, lines 27 to 30 would seem to say that ploughed and non-ploughed systems are similar in terms of N supply to plants, is that what you mean? Clarification may be needed.

These sentences were rewritten in the lines 48-51.

**Reviewer 3**

1. Specific comments One key concern is that N mineralization is measured under laboratory conditions and then corrected to field conditions, via a solely temperaturedependent Q10 equation (L112-114). It is well known that the simple Q10 relationship does not hold under realistic soil conditions, since temperature is not the only limiting factor. Soil moisture, substrate availability, etc also strongly co-determine the biogeochemical process rates in situ (see e.g. Davidson & Jansses 2006 Nature 440: 165-173 for SOM decomp). Therefore, I do not believe that the authors can capture realistic N mineralization rates in their field. I think this paper needs a thorough validation of this relationship.

The lab incubation method and the in situ method are the most common methods used in research to investigate the soil N mineralization rate. But, both methods have their own limitations (Hanselaman et al., 2004; Wienhold, 2007). So, obtaining the absolute real value is virtually impossible.

In the revision, we used the inorganic nitrogen content measured in fresh soil sampled from the field every month instead of the amount of mineralized N obtained through lab incubation to indicate the status of soil N under different tillage systems during soybean growth. Determination of inorganic N by direct monthly field measurements integrates the contribution of all of the factors mentioned by the reviewer affecting N mineralization rate. We want to emphasize that our core objective was to make a comparison among different tillage systems, not to obtain absolute values of N mineralization rates. In the revision, we reworked to objective to clarify that the focus was on comparing the tillage systems.

Since the same test method was used for all tillage systems, errors or biases caused by the test method would be the same for samples collected from different tillage systems.

2. Similarly, I am highly critical of the way the authors attribute N mineralization contributions from different soil biota groups. They use a series of equations from other authors to transform soil biota abundances into process rates (e.g. L170-L176, L177-188, L198-202). Mostly these steps seem to be based on Rashid et al 2014. These steps form the heart of their study. For instance, the conclusion that conservation tillage promotes N min (L21-23), hinges on these equations that all assume that more soil biota lead to more N min. The same goes for the relative contributions of soil biotic groups to total N mineralization (L25-27). The parameter estimates (e.g. Q10 of 3, L116) used come from different systems in other countries, while it is know that N cycling processes are highly heterogeneous in space and time. I am therefore sceptical that the same relations and the same parameter estimates will hold in the system studied by the authors. In fact even in the source paper, Rashid et al 2014, the ecologicalproduction model is an improvement over the standard government rules, but still there is considerable error in the estimates (87-120% of observed N min rates) on the fields they studied. So I think the authors have to spend much more effort on convincing me and other readers that using these equations leads to valid inferences about this particular system. To be honest, as an empiricist, I think that to only realistic way to get to these questions is to use isotopic tracers in the field plots. However, what would help is if 1) we had realistic data on N min rates in the actual plots, and 2) the summed N contributions over the soil biota would have a strong predictive relationship with these independent field data. As it stands such a field validation is totally missing, which makes the study unconvincing.

Researchers have used theoretical methods to quantify the elemental energy flux of soil food webs for more than thirty years. The parameters, such as assimilation efficiency, the ratio of C:N of predator or prey, and feeding preference and so on, used in this method were almost constant over the past thirty years. The classic literature is de Ruiter et al. (1993), Didden et al. (1994) and Hunt et al. (1987), and the recent literature of Andrés et al. (2016, Soil Biology and Biochemistry), de Vries et al. (2013, PNANS) and Schwarz et al. (2017, Nature Climate Change) also used this method to explore the C or N flow through soil food webs in the grassland ecosystem of America, agroecosystem of Europe and the forest ecosystem of America. The method is well established and accepted by researchers. So far, as far as we know, there is no research using this theoretical method to quantify the energy flux of the soil food web in Asia or China.

In the revised manuscript, we re-calculated the N mineralization of soil organisms according to Ruiter et al. (1993), see line 217-233 for details. This method does not require the use of Q10. And we also discussed the influence of physiological parameters that are required for the calculation of N mineralization in the lines 329-342. As discussed in our response to comment 1 above, our objective was to compare tillage systems, not to obtain absolute values of N mineralization rates of the various soil biota. This objective was clarified in the revision.

We agree that the use of isotopic tracers would be a good way to obtain the actual N mineralization data independent of assumptions. However, that is a major study in itself and is well beyond the scope of this manuscript. We think that our findings will provide good background material for further studies in isotope tracing and we mentioned this in the line 441-448 in the revision.

3. Data were missing in some months for nematode data and linear interpolation was used to fill these data gaps (L129). I find this a risky approach, especially since nematode population dynamics within season are non-linear, see e.g. the data in Rashid et al, but also other sources. I think the authors also need to show that their conclusions hold if the only work with the months where they have data on all soil groups.

The nematode populations for non-sampled months were estimated by linear interpolation between adjacent sampling dates. Ideally, more frequent sampling would be done, but as with most research projects, our resources were limited. This method (linear interpolation) is usually used in the literature (Didden et al., 1994; Berg et al., 2001; Zhang et al., 2019), which assumes that there is a linear course in biomass or abundance of soil organisms between sampling dates. This method can not track the short term trends of nematode population changes, but can yield a reasonably accurate mean value during the studied period.

Line 159-160: we rewrote these sentences to make it clear for readers.

4. The authors use the ratios of (calculated) mineral N delivery in the conservation tillage (ridge, and no tillage) to conventional tillage in their main figures. However, ratios are biased (e.g. Jasienski & Bazzaz 1999 Oikos 84: 321-326); a log(Treat/Control) has better statistical properties (Brinkman et al 2010 J Ecol 98: 1063–1073). Even better however would be if the main analyses and figures are directly based on the data from the three treatments directly, this approach would even give you a bit more statistical power. In that sense I find the supplementary figures to be much clearer.

Thank you for your suggestion. The tables and figures were reorganized in the revised manuscript. We used max-min normalization to compensate for the wide difference in ranges spanned by the various parameters, and $\ln(x+1)$ transformation to improve the normality of the data prior to statistical analysis. The detailed analysis information was presented in the line 246-266. In the revision, we deleted the calculations and figures on the various N delivery ratios for the tillage systems.

5. In general, I find that the writing is a bit to colloquial in tone and imprecise in many places. See some examples below. Also I find that the presentation of the energy channels to be a bit overstated, there have been many findings of cross-feeding across these 'channels', and really I think we need to adopt a network view of the soil community and its links to biogeochemical processes.

The soil food webs were rebuilt in the revised manuscript. Sensitivity analysis was conducted to test the influence of the uncertainty of the assignment for omnivorous collembolans on the result of N mineralization. All ambiguous results were deleted, and the discussion was rewritten to obtain a concise and logical conclusion.

6. Minor comments - L44: what do you mean with 'special species'? - L51: what are weak root infections - L55: what do you mean by capacity? Use of substrates? Process
rates? - L60: I would not use the word conquer here, maybe mediate? - L61: adverse effects on what? - L66: rich in what sense - L68: what is stratrified and in what way? -
L80: based on M&M I believe its 14 years, not 15. - L83: what do you mean coupling?
How will you quantify that coupling? - L85: it is a bit unclear what you mean by multiple spatial interactions in this hypothesis. How will you test this? - L94: how big were the plots? - L100: what was done with the maize residue?

These inappropriate points in the part of introduction were rewritten, please see Line 67-91 in the revised manuscript. And the hypothesis was also rewritten in the line 108-111 to avoid ambiguous and unclear words.

[revised manuscript text omitted]

---

## Referee Report (RR1)

[referee-annotated manuscript omitted]

---

## Author Response (AR2)

Dr. Elizabeth Bach
Topical Editor
SOIL

Dear Editor,

Re: Manuscript soil-2020-2 "Relationships between N mineralization of soil organisms and soybean yield in conservation tillage systems" Shixiu Zhang et al.

Thank you very much for your careful review and constructive suggestions with regard to our manuscript. We sincerely appreciate the reviewer's and Editor's thorough reviews and helpful suggestions. I am sending here one copy of our revised manuscript, with the revised portion marked in red, and a file with a revised appendix.

The responses to the reviewer's comments are listed below. The content marked in red in the revised manuscript is for the convenience of tracking the modification according to the reviewers' suggestion. But, other contents, such as abstract, introduction, results, discussion and conclusion, etc., were also revised intensively.

We believe that we have addressed all of the reviewers' comments and that manuscript has been improved satisfactorily. We hope it will meet your approval.

Yours sincerely,

Dr. Shixiu Zhang

Comments from the reviewers are in normal font and our responses are marked in blue. The line references in our responses refer to the line numbers in the revised manuscript.

**Reviewer 1**

1. I think the manuscript can benefit from a discussion (either in the Introduction or Methods) on the utility and limitations of energetic food web modeling. Indeed, these methods have been used for several decades now, but they are not mainstream, and many readers interested in this study will likely be less familiar. From my understanding, energetic food webs are highly parameterized theoretical models that are meant to give crude estimates of C and N cycling rates through food webs; the results are not meant to be interpreted as absolute. Addressing these expectations/limitations upfront should help eliminate the confusion of why so many parameter values are derived from different systems while also elucidating the usefulness of this method.
Thank you for your suggestion. We added the description regarding the utility and limitations of food web energetic modeling approach in the part of Introduction (Line 130-135) and Methods (Line 283-287) to help readers better understand this approach.

2. Furthermore, there should be a figure that illustrates the energetic food web models (similar to Figure S1), ideally showing all three tillage treatments so that they can be compared visually (see Pressler et al. 2018 for example). The line size of each pathway should indicate the strength of the interaction between food web compartments (the 'igraph' package in R is one way to do this). I would also suggest eliminating (or reducing) Table 3 that shows the same information or placing that information in the appendix.
Referring to the example presented in the study of Pressler et al. (2018), a vividly visible diagram Figure 2 showing the N flux across multi-trophic feeding guilds among different tillage system was added in the main text. Table 3 was moved to the supplementary file and renumbered as the Table S7.

3. I propose adding a separate section for the Sensitivity Analysis under the 'Statistical Analysis' heading in the Methods. Explicitly state what was done and if the results are consistent when using different feeding preferences and/or different food web configurations (see Koltz et al. 2018 for example) -- simply stating that "a sensitivity analysis was performed by re-assigning omnivorous collembolans into fungivores and herbivores (50% each)" is not sufficient. A formal sensitivity analysis would help evaluate the robustness of the results in this ecological context. There should be text explaining how it was performed in the Methods, reported in the Results, and discussed in the Discussion.
Thank you for your suggestion. After carefully reading the reports of Koltz et al. (2018) and other studies (Barnes et al., 2014; Carrillo et al., 2016; Schwarz et al., 2017; Zhang et al., 2018) that also conducted Sensitivity Analysis to test the robustness of the data, we adhered to our approach but made some modifications to thoroughly test the changes in all functional feeding guilds. The presented Sensitivity Analysis was based on Barnes et al. (2014) and more detailed information was added in the part of 'Statistical Analyses'

(Line 302-308), 'Results' (Line 358-366) and 'Discussion' (Line 413-438).

The primary reason for we did not refer to Koltz et al. (2018) to perform the Sensitivity Analysis is that we think their assumption is impractical for our study. Koltz et al. (2018) conducted the Sensitivity Analysis based on the assumption of incomplete food webs (feeding groups removed from the network at one time) and creating an additional food web (feeding groups have no specified feeding preferences).

Although this assumption can ensure a formal sensitivity analysis, it is not suitable for our study because almost all feeding guilds, except omnivorous collembolans, have explicit feeding preference and position in the detritus food web network. Therefore, we did the Sensitivity Analysis according to Barnes et al. (2014) which is more reasonable for field conditions. Furthermore, we made a modification to not only show the changes in the whole food web, but also to show the changes in each feeding guild to fully understand the consequences of variation in feeding preferences of omnivorous collembolans. This can provide more useful, practical and realistic information (Line 424-438) than that method of Koltz et al. (2018).

4. The various types of tillage treatments need to be clearly defined in both the Introduction and Methods section. What is the difference between conservation tillage vs. conventional tillage? Which one of the experiment treatments is considered to be conservation tillage? Is the NT the control treatment? There appears to be a disconnect between the terminology used in the Introduction and Methods; the terms should be consistent and clear throughout the manuscript. Not all readers will be familiar with agricultural practices.

More detailed information about the difference between conventional tillage and conservation tillage (including reduced tillage and no tillage) was added in the part of Introduction (Line 90-93). The terms and abbreviations of different tillage practices have been checked and used consistently throughout the whole text in the revised manuscript.

5. The hypotheses in the last paragraph of the Introduction need to be developed further. Why do you expect there to be more N release from conservation tillage from conventional tillage? Why would soil depth be pertinent for determining the influence of soil organisms regarding N mineralization?

The hypotheses were rewritten in the line 121-125 to make them clear to readers.

6. Overall, the Discussion is long and disorganized. I suggest starting with a short summary of the results, making sure to highlight the key new insights, followed by a discussion of if the hypotheses were supported or not. The middle paragraphs should be used to explain what the findings mean within a broader context, drawing on support (or contrasting support) from the literature. I will reserve the section on study limitations and commentary on the sensitivity analyses for the section right before the Conclusions paragraph.

Thank you for your suggestion. We reorganized discussion in the Line 399-521, and used the writing technique you mentioned to ensure that readers can easily find the

summary results in the opening paragraph of each subtitle and whether these results support our hypothesis.

To make it more logical and smoother for readers to follow, the discussion was partitioned into the following subsections: 4.1 performance of modeling N mineralization within the food web (line 413-438), 4.2 tillage effects on the N mineralization within the food web (Line 441-458), and 4.3 relations between N mineralization within the food web and soybean yield (Line 461-521).

7. The Abstract is confusing. I'm not sure what results are from the energetic modeling, N mineralization measurements, or other associated analyses. For example, in Lines 46-48, is this the energetic modeling result, or is this from the biomass estimates of the trophic guilds? In general, the Abstract needs to be summarized more effectively, as the current take-home message is unclear.

The Abstract was rewritten in lines 30-57.

8. The way the experiment is described is somewhat misleading, as at first read it appears that the experiment and its sampling have been going on since 2001, which was not the case. To make the experimental design clearer, I suggest stating the time interval of the tillage treatments (e.g., 2001 – 2015) and mention that this study only sampled for N mineralization and soil organisms in 2015.

The description of the time over which this study was conducted was revised and it was highlighted in the parts of 'Title' (Line 8), 'Abstract' (Line 36), 'Introduction' (Line 128) and 'Methods' (Line 176) to make it clear for readers.

9. The extraction of 120 hours at room temperature is not standard for Berlese-Tullgren funnel methods. What is the justification here, and how do you know that all arthropods were extracted if there wasn't a heat gradient?

We used the modified high-gradient Tullgren funnels (Crossley and Blair, 1991) method to extract all microarthropods. This method uses small light bulbs as a heat source above the funnels and yielded a gradient of temperature and moisture along the extractor to ensure that microarthropods can be collected at the bottom of the extractor. Many reports in the literature (Kardol et al., 2011; Risch et al., 2018; Soong et al., 2016) used this method to study the microarthropod community; Google scholar shows 181 citations of the Crossley and Blair (1991) paper which is a good indication that the method is widely used.

10. Although there is information on the various groups of soil fauna sampled in Table S1, it would have useful to include the average densities and standard errors for each taxonomic unit according to tillage treatment. Soil core extractions will usually capture springtails, mites, and nematodes – which of these groups and functional guilds were most abundant?

The abundance of soil fauna was added in the supplementary file as Table S4.

11. The assumptions of the model are acceptable and similar to previously published

models (e.g., Schwarz et al. 2017, Koltz et al. 2018, Pressler et al. 2018). However, because some of the literature values are from very different regions and study systems, the authors should explain why these parameter values are appropriate for this study. Obviously, it would've been nice to measure and confirm each parameter value, but that is usually impossible to accomplish for these types of models.

The explanation about how the parameter values were chosen was added in the line 283-287.

12. Lines 221-223: I would recommend stating if the N mineralization equations are modified from de Ruiter et al. (1993), and if so, how?

Line 269-273: There was no modification for these equations; in the revision, we clarified that we used the Equations from de Ruiter et al. (1993).

13. Lines 305-314: The Results don't do an adequate job in covering how the food web modeling relates to the crop yield. Maybe I missed this, but is there a way to loosely compare the energetic food web modeling to the crop yield data?

The relations between the mineralized N within the soil food web and the soybean yield was explored by the forward stepwise multiple linear regression (MLR) model. We rewrote the sentences in the line 318-320 and line 388 and 393 to make it clear to reader.

**Reviewer 2**

1. Overall comments: One year of data is simply not enough to draw the conclusions you have done here. This is a long-term study so why not include historical yield data? There were no crop yield differences and the mineral N concentrations, in 0-15 cm depth, were contrary to expectations. The estimations for N cycling within the soil food web are therefore rendered difficult to interpret or meaningless. If only one year of data is presented, without any data on growing conditions, could it not be simply that N mineralization was controlled more by soil water content and temperature?

Our study was focused on investigating how the whole soil organism communities regulate nutrient cycle impacting crop growth not on the variation in crop yield among different tillage practices. Furthermore, it is impractical to monitor the changes in the whole soil organism communities in every year. Although the samples were only taken within one year during the soybean growing season, the sampling time was 14 years after initiation of long-term (continuous) conservation tillage practice. It is well documented (Six et al., 2004) that soil environment reaches a new stable equilibrium after a long-term (>10 years) application of conservation tillage. Therefore, our samples are representative and can obtain reliable results to understand the relations between mineralized N delivered by soil food web and soybean yield after long-term application of conservation tillage system.

There was indeed no statistical analysis for the soybean yield among different tillage practice, but there was a lower ($P = 0.027$) soil mineral N pool of 0-15 cm in NT than in CT (Table 1). This is the key point and poses the question of why the low soil mineral N pool in NT supports the soybean yield comparable to CT? Therefore, we proposed

that NT has a larger mineralizable N pool that was regulated by soil organisms than CT, and this was supported by the presented results in our study. A concise summary for the relations between soybean yield and soil mineral N pool and mineralizable N pool was added in line 399-410.

We agree that in addition to soil organisms, soil moisture and temperature also influence N mineralization, but the underlying process of N mineralization is primarily governed by the activity of soil organisms which are live in the water film. Additionally, our study was focused on investigating the effect of tillage on soil N mineralized by organisms as opposed to determining absolute quantities of soil N mineralized.

The historical yield variation at the same experimental site has been reported in the studies of Fan et al. (2012) and Zhang et al. (2015).

2. Ln 6, replace "Relationships" with "Relations", and remove "of soil organisms". Use "relation" throughout, where applicable.
   Line 7, 'relationships' was replaced by 'relations', and "of soil organisms" was deleted. In the whole revised manuscript, the 'relationships' was replaced by 'relations'.

3. Ln 7, as it stands, the title is rather broad. I would suggest making it more specific. For example, you could include where the study was conducted as part of the title.
   Line 7, the title has been rewritten.

4. Ln 33, suggest changing "promoting" to "to promote"
   The Abstract has been rewritten, so "promoting" was deleted.

5. Ln 34, again, in the N being mineralized exclusively from soil organisms themselves? You could say "by soil organisms", or just "...N mineralization and soybean ..." because the process is assumed to be biologically driven/mediated.
   Thank you for your suggestion, we changed the preposition from 'of' to 'by' to relate N mineralization and soil organisms.

6. Ln 41, as you plowed to 20 cm but only measured the N release to 15 cm you cannot say "thoughout the plow layer". Rephrase.
   'throughout the plow layer' was replaced by 'the entire soil layer (0-15 cm)' in the whole revised manuscript.

7. Ln 44, it is unclear what a "plant channel" is and how it mineralizes N.
   To make it clear to readers, 'plant channel' was replaced by 'root pathway' and its definition was given in the line 295. The role of how soil organisms in root pathway mineralize N was given in the part of introduction (Line 72-80) and in the part of discussion (Line 504-521).

8. Ln 42-46, the logic of this sentence is hard to follow. It is also a run-on sentence. Soybean yield was related to N mineralization, though fungal, plant, and bacterial channels (?) and this somehow demonstrates the role of spatial variability? Remove

"demonstrating the role of spatial variability of soil organisms in linking N mineralization to plant growth".

These sentences were rewritten in the line 50-54.

9. Ln 47, the sentence would need to define what "energy channel" you are referring to. Be specific.

The sentence was reworked in the line 49.

10. Ln 51, change "optimal" to "maximum". Optimal is a subjective term and you have not defined it here.

Abstract has been rewritten, so "optimal" was deleted.

11. Ln 60, include citation(s) that state this.

The citation was added in Line 66.

12. Ln 61-63, you have omitted portions of the soil N cycle. Simply because N recovery in plants from fertilizer application is not 100% doesn't mean that all is lost. Immobilization could be a factor as well. Or that pools of inorganic N in soil build up. So, include all possible loss pathways in your statement.

The possible loss pathways of inorganic N were added in line 69.

13. Ln 63-66: How does one exploit the role of soil organisms? Make your meaning explicit.

These sentences have been reworked in the line 70-71.

14. Ln 73: In what form is the excreted N? Is it plant available?

Line 80: the sentence was replaced by 'the excess N is excreted into the soil ammonium ($NH_4^+$) pool' to make it clear to readers.

15. Ln 79: What accounts for the rest of the N mineralized? Don't leave the reader wondering here.

Line 82: It is impossible to include every trophic group in the soil to simulate the N mineralization, and thus the simulated N amount released from the predation of soil organisms only accounts for 30%-80% of the annual N mineralization. Therefore, we attributed the remainder of the mineralized N to the uninvestigated trophic groups. This was clarified in the revision.

16. Ln 80-82: I think that this statement is too bold. It would seem that your citation refers to the second part of the sentence, i.e., conservation tillage can promote richness and abundance of soil organisms. You need to cite the source of your statement that conservation tillage is "one of the most efficient practices to maintain optimal productivity". Presumably you mean agricultural practices. In what way would it be efficient? No-till systems can take several years to "settle", so chemical fertilizers are likely more efficient in that sense. Also, what is "optimal"? Maximum?

These sentences were rewritten in the line 90-99.

**17.** Ln 97, please clarify what you mean by "benefit". Who or what benefits here? The producer? But in what sense?
The sentence was rewritten in the line 93-94.

**18.** Ln 99, what changes are you referring to? Be specific and start the sentence with the subject.
These sentences were rewritten in the line 103-107. So, 'changes' was deleted.

**19.** Ln 102-104: I would expect mention of soybean from the title of the manuscript. In the title you say "soybean yield", so is it just one crop or all?
We changed 'plant yield' to 'soybean yield' in the title (Line 8) and in the objective (Line 119-121).

**20.** Ln 108, specify the time frame? Over a rotation? Over the entire length of the study? Season? Or from week to week?
The time frame was added in the 'Title' (Line 8), 'Abstract' (Line 36), 'Introduction' (Line 128) and 'Methods' (Line 176) to make it clear for readers.

**21.** Ln 109, I think that you should be precise about what the "key role" is that you are looking at. How will you judge if a set of organisms plays a "key role" at one depth and not at another?
The hypotheses were rewritten in the line 121-125. So, the 'key role' was deleted.

**22.** Ln 119: Missing article at the beginning of this sentence.
Agreed, the article was missing. We added 'The' at the beginning of the sentence in line 149 so it now reads: "The long-term tillage experiment was established …."

23. Ln 124: Please choose one system of spelling and be consistent. Here you mix British and American.
Line 155: 'mouldboard plowing' was replaced by 'moldboard plowing' and consistent throughout the whole revised manuscript.

**24.** Ln 126: Could you please state what the common practice would be in the study area? How common is seeding maize and soybean into ridges?
The detailed information about the local common tillage practice was added in the part of instruction in the line 108-112. Traditional practice is to create ridges to promote soil warming, and seed maize and soybean into the ridges.

**25.** Ln 124-128, since tillage systems are the focus of your research you should refer to the specific instruments that were used. Make and model. As you would for analytical equipment. This is needed to judge the relative intensities of the systems. The reader likely won't know what a "modified lister and scrubber" looks like or who intense the soil modifications would be.

The model and the produced company of no-till planter was added in Line 160. The other instruments that used in different tillage practices had no specific model and company.

26. Ln128- 132, Please provide an explanation as to why residues were replaced. Is this meant to mimic the amount of residue left in such systems under commercial circumstances? Are you not biasing the results by not incorporating the residues in the CT? The lack of inputs in the plough layer would presumably have an influence on the soil biology.

Preventing the water and wind erosion in winter and early spring is the reason for why residues were replaced on the soil surface after fall moldboard plowing in CT, and this reason was added in the line 160-162. Additionally, these residues laid on the soil surface in CT plots were mixed with the plow layer in the following spring tillage, so there was no bias caused by unmixed residues. This was clarified in the revision (line 164-166).

27. Ln 135-136, please provide all relevant information on experimental design here so the reader doesn't have to refer to a separate article. This information is important to the current work.

Detailed information about the experimental design was added in the line 152-154, line 160-162 and line 164-172.

28. Ln 137, specify what years the data were taken in.

The specific sampling year was given in the line 176.

29. Ln 152, you should also include weather and climate data. How representative of the historical precedent were the times when you measured?

The mean precipitation during the growing season of 2015 and of the past 10 years (2004-2014) were added in the line 176-179.

30. Ln 149-152, specify in what years you measured crop yield.

The specific year that measured crop yield was added in the line 194.

31. Ln 217-219, provide a justification as to why the assumption of equilibrium is valid. If C:N shifts, SOM buildings or decreases then it would seem that this assumption is not valid?

The mass-balance assumption of soil food web energetic model approach, that is the energy flowing into the biomass of a group is equal to the energy flowing out through natural death and predation, is the basal principal and necessary requirement for simulation the mineralized N delivered by soil organisms; the assumption is widely used in this type of research. This assumption is based on the biomass, metabolic constant, and allometric growth of soil organisms and not based on the soil environment variance (Barnes et al., 2018). So, this assumption can be applied in a range of soil environments that have different C:N ratios and SOM contents.

**32.** Ln 246, stating it in this way would seem to say that you transformed the data prior to testing for assumptions. This would be an incorrect approach.
These sentences were rewritten in the line 309-311 to make it clear that the data were first checked for normality and homogeneity of variances, and if necessary, transformed to meet the underlying assumptions for ANOVA.

**33.** Ln 255, please explain what you mean by "channel".
The sentence was reworked in the line 319 to make it clear to reader.

**34.** Ln 265-266, please properly cite all R packages used.
The R packages that used in this study were carefully checked and presented in the line 328-330.

**35.** Ln 270-275, please specify the form of N that you are refering to. nitrate, ammoniumn, or total?
In this study, soil mineral N was obtained by the sum of $NO_3^-$ and $NH_4^+$. This was clarified in the Methods section in line 200.

**36.** Ln 271, This statement would appear to incorrect according to Table 1. RT > CT but NT is not different than either, at 0-5 cm.
These sentences were rewritten in the line 335-338.

**37.** Ln 272, it is probably not appropriate to call the 5-15 cm depth the "deep layer" as you have measured N content at an even deeper layer.
Line 336, 'deeper layer' was deleted.

**38.** Ln 273, this trend is not significant, so in fact, your analysis shows that CT > RT = NT.
Although there was no significant difference between RT and NT, the magnitude of soil mineral N value of 0-15 cm was decreased in the order of CT (21.68) > RT (20.71) > NT (18.98) (Table 1).

**39.** Ln 323, a description of how this was done should be in the Materials and Methods section.
The sensitivity analysis was added in the part of statistical analyses in the line 302-308 of the Methods section.

**40.** Ln 367-369, Rephrase this statement, it is misleading. CT>NT, but RT was not different than either.
These sentences were rewritten in the line 402-403. Additionally, the order described the magnitude of soil mineral N and soybean yield and did not refer to the results of statistical analysis.

**41.** Ln 369-370, Again, it is misleading to insinuate there were differences when in fact,

Table 1 states that there were no signficant differences in crop yield.

As statement in the above, the order described the magnitude of soybean yield and did not refer to the results of statistical analysis.

**42.** Ln 451, it would be more appropriate to say that "we estimate" since you did not measure this directly.

Conclusion was rewritten in the line 524-535, so 'showed' was deleted.

**43.** Ln 456, never were the mineral nitrate concentrations higher in NT than CT, nor was crop yield higher in NT than CT, so this statement is misleading. Please re-phrase.

These sentences were removed in the revised manuscript as the part of conclusion was rewritten.

**44.** Table 1, when were these soil samples taken? At the end of the season? If there were no yield differences, in one year at least, but there was a difference in mineral N left after harvest (?).

The detailed information about sampled taken time was added in the line 176.

**45.** Ln 635, please remind the reader where these data came from.

These data were calculated using the equations according to de Ruiter et al. (1993) and detailed calculation process was presented in line 247-299.

**46.** Ln 637, please rephrase without using repetitive terminology.

These terms were reworked at the bottom of Table S7 as the Table 3 was moved to the supplementary file according to the suggestion of reviewer 1.

**References**

1.  Barnes, A.D., Jochum, M., Lefcheck, J.S., Eisenhauer, N., Scherber, C., O'Connor, M.I., de Ruiter, P. and Brose, U.: Energy flux: the link between multitrophic biodiversity and ecosystem functioning, Trends in Ecology & Evolution, 33, 186-197, doi: 10.1016/j.tree.2017.12.007, 2018.

2.  Barnes, A.D., Jochum, M., Mumme, S., Haneda, N.F., Farajallah, A., Widarto, T.H. and Brose, U.: Consequences of tropical land use for multitrophic biodiversity and ecosystem functioning, Nature Communication, 5, 5351, doi: 10.1038/ncomms6351, 2014.

3.  Carrillo, Y., Ball, B.A. and Molina, M.: Stoichiometric linkages between plant litter, trophic interactions and nitrogen mineralization across the litter - soil interface, Soil Biology & Biochemistry, 92, 102-110, doi: 10.1016/j.soilbio.2015.10.001, 2016.

4.  Crossley, D.A. and Blair, J.M.: A high-efficiency, low-technology tullgren-type extractor for soil microarthropods, Agriculture, Ecosystems & Environment, 34, 187–192, doi: 10.1016/0167-8809(91)90104-6, 1991.

5.  de Ruiter, P.C., van Veen, J.A., Moore, J.C. Brussaard, M.L. and Hunt, H.W.: Calculation of nitrogen mineralization in soil food webs, Plant & Soil, 157, 263-273, doi: 10.1007/BF00011055,

1993.

6. Fan, R.Q., Zhang, X.P., Liang, A.Z., Shi, X.H., Chen, X.W., Bao, K.S., Yang, X.M. and Jia, S.X.: Tillage and rotation effects on crop yield and profitability on a Black soil in northeast China, Can. J. Soil Sci. 92, 463-470, doi: 10.1139/CJSS2010-020, 2012. .

7. Kardol, P., Reynolds, W.N., Norby, R. and Classen, A.T.: Climate change effects on soil microarthropod abundance and community structure, Applied Soil Ecology, 47, 37-44, doi: 10.1016/j.apsoil.2010.11.001, 2011.

8. Koltz, A.M., Asmus, A., Gough, L., Pressler, Y. and Moore, J.C.: The detritus-based microbial-invertebrate food web contributes disproportionately to carbon and nitrogen cycling in the Arctic, Polar Biology, 41, 1531-1545, doi: 10.1007/s00300-017-2201-5, 2018.

9. Pressler, Y., Foster, E.J., Moore, J.C. and Cotrufo, M.F.: Coupled biochar amendment and limited irrigation strategies do not affect a degraded soil food web in a maize agroecosystem, compared to the native grassland, Global Change Biology Bioenergy, 9, 1344-1355, doi: 10.1111/gcbb.12429, 2017.

10. Risch, A.C., Ochoa-Hueso, R., van der Putten, W.H. Bump, J.K., Busse, M.D., Frey, B., Gwiazdowicz, D.J., Page-Dumroese, D.S., Vandegehuchte, M.L., Zimmermann, S. and Schütz, M.: Size-dependent loss of aboveground animals differentially affects grassland ecosystem coupling and functions, Nature Communication, 9, 3684, doi: 10.1038/s41467-018-06105-4, 2018.

11. Schwarz, B., Barnes, A.D., Thakur, M.P., Brose, U., Ciobanu, M., Reich, P.B., Rich, R.L., Rosenbaum, B., Stefanski, A and Eisenhauer, N.: Warming alters energetic structure and function but not resilience of soil food webs, Nature Climate Change, 7, 895-900, doi: 10.1038/s41558-017-0002-z, 2017.

12. Soong, J.L., Vandegehuchte, M.L., Horton, A.J., Nielsen, U.N., Denef, K., Shaw, E.A., de Tomasel, C.M., Parton, W., Wall, D.H. and Cotrufo, M.F.: Soil microarthropods support ecosystem productivity and soil C accrual: Evidence from a litter decomposition study in the tallgrass prairie, Soil Biology & Biochemistry, 92, 230-238, doi: 10.1016/j.soilbio.2015.10.014, 2016.

13. Zhang, S.X., Chen X.W., Jia S.X., Liang A.Z., Zhang X.P., Yang X.M., Wei S.C., Sun B.J., Huang D.D. and Zhou G.Y.: The potential mechanism of long-term conservation tillage effects on maize yield in the black soil of Northeast China, Soil & Tillage Research, 154, 84-90, doi: 10.1016/j.still.2015.06.002, 2015.

14. Zhang, Z.L., Xiao, J., Yuan, Y.S., Zhao, C.Z., Liu, Q. and Yin, H.J.: Mycelium- and root-derived C inputs differ in their impacts on soil organic C pools and decomposition in forests, Soil Biology & Biochemistry, 123, 257-265, doi:    , 2018.

---

## Author Response (AR3)

Dr. Elizabeth Bach
Topical Editor
SOIL

Dear Editor,

Re: Manuscript soil-2020-2 "Relations between mineralized N delivered by soil food web and soybean yield after long-term application of conservation tillage system in a black soil of Northeast China (soil-2020-2)" Shixiu Zhang et al.

Thank you very much for your careful review and constructive suggestions with regard to our manuscript. We sincerely appreciate the Editor's and Reviewer's thorough reviews and helpful suggestions.

I am sending here one copy of our revised manuscript, with the revised portion marked in bright red and double lines for the convenience of tracking the modification. The responses to the reviewer's and editor's comments are listed as appendix behind this letter. Please note that the title above is that of the original submission. As part of the revision, we have changed the title to "Complex soil food-web enhances the association between N mineralization and soybean yield: A model study from long-term application of conservation tillage system in a black soil of Northeast China".

We believe that we have addressed all of the Editor's and Reviewer's comments and that manuscript has been improved satisfactorily. We hope it will meet your approval.

Yours sincerely,

Dr. Shixiu Zhang

Comments from the Editor and Reviewer are in black and our responses are marked in blue. The line references in our responses refer to the line numbers in the revised manuscript.

**Editor**

**1.** Must address caveats pertaining to 1 year of data.

Thanks for your suggestion. We added the caveats of 1 year of data collection in the lines 415-419 of the revised manuscript.

**2.** Must clearly present as a modeling study that sets-up hypotheses for future studies to test, not as empirical data. The final sentence of the conclusion (lines 530-535) does this, but it needs to be evident in the abstract, the results, and early in the discussion.

In order to highlight the goal of 'present study is acted as a modelling study not as empirical data', we revised some places in the abstract, the results, and early in the discussion according to the suggestions. The detailed revised places were in lines 55-59 (abstract), 357, 359, 390 (results), 415-419 and 546-548 (discussion) of the revised manuscript. Furthermore, we also revised the title (lines 7-10) to make it more in line with this goal.

There is lingering confusion about the relationship of these results to soybean yield. The title and paper clearly tie the soil food web modeling to soybean yield, yet in response to reviewer 2, it is stated "Our study was focused on investigating how the whole soil organism communities regulate nutrient cycle impacting crop growth not on the variation in crop yield among different tillage practices." There is a subtle difference in soil community contribution to crop growth vs. tillage differences in crop yield. Crop growth was not measured in this study, yield was, and the paper repeatedly ties the results back to soy yield and tillage practice.

I am very sorry for the misunderstanding caused by the unclear expression. The focus of this study is indeed to investigate the relationship between N mineralization mediated by soil organisms and soybean yield after long-term application of conservation tillage. Although the N mineralization within soil food web and soybean yield were both presented in this study, more attention was put on the simulation of N mineralization within the whole food web rather than put on the soybean yield. This is primarily because that we want to test the importance of soil food complexity in affecting crop yield through mediating nutrient cycling.

We agree that 'crop growth' and 'crop yield' refer to different content. So, in order to be clear to readers and avoid any confusion, the term 'crop growth' was deleted throughout the revised manuscript.

**Referee 4**

3. Line 32: Change to "achieve a new equilibrium in the soil environment"

Line 32: the sentence was revised.

4. Lines 34-35: Change to "in such a situation" ... "how the soil community regulates nutrient cycling impacting crop growth is not well documented"
Lines 34-35: the sentence was revised according to reviewer's suggestion.

5. Lines 36-37: Delete "continuous application"
Line 37: 'continuous application' was deleted.

6. Line 39: Delete "pool"
Line 39: 'pool' was deleted.

7. Line 40: Change to "plant growth will vary"
Line 41: 'will' was added.

8. Lines 44-45: Change to "using energetic food web modeling."
Lines 45-46: 'using the food web energetic model' was replaced by 'using energetic food web modeling'.

9. Line 47: Change to "A similar trend…"
Line 48: 'Similar trend' was replaced by 'A similar trend…'.

10. Line 56: Should be "potential mineralizable N pool"
Lines 55-57: this sentence was rewritten, so 'potential mineralizable N pool' was deleted in the revised manuscript.

11. Lines 56-57: "which is a cornerstone for conservation tillage system to achieve the sustainable crop productivity" – This sentence fragment needs revising; I'm not sure what you mean here.
Lines 55-59: these sentences were rewritten in the revised manuscript.

12. Line 87: Should be "potential mineralizable N pool"
Lines 90: 'the potentially mineralizable N pool' was replaced by 'potential mineralizable N pool'. The whole context was thoroughly checked, thereby the following places in lines 467 and 535 were also revised in the manuscript.

13. Line 95: Change to "when the soil environment"
Lines 98-99: 'the' was added.

14. Line 97: Change to "situations"
Line 100: 'situation' was replaced by 'situations'.

15. Line 105-107: This sentence needs to be revised; hard to understand
Lines108-110: we rewrote this sentence to make it clear to readers.

16. Line 123: Change to "plant growth will vary"
Line 126: 'will' was added.

17. Line 161: Change to "prevent water and wind erosion"
Line 164-165: 'prevent the water and wind' was replaced by 'prevent water and wind erosion'.

18. Line 179-182: I think instead the authors should acknowledge the limitations of data collected in only one year.
Thanks for your suggestion. We acknowledged the limitations of one-year data in lines 415-419 of the revised manuscript.

19. Line 193: Need to close out the sentence with a closing bracket
Line 195: the closing bracket was added.

20. Line 206: Change to "The microbial community"
Line 208: 'The' was added.

21. Line 232-234: The arthropods are not extracted at room temperate if there is a heat gradient from the lights. I would delete the phrase "at room temperature" because it is confusing and not the standard language for describing this method.
Line 236: 'at room temperature' was deleted.

22. Line 526: Should be "potential mineralizable N pool"
Line 540: 'potentially' was replaced by 'potential'; this terminology has been carefully checked throughout the context to make sure that 'potential mineralizable N pool' was consistently used in the revised manuscript.

In addition to the above revisions, we have also made minor revisions where there were grammatical problems or unclear sentences. All revisions were marked with bright red in the revised manuscript.